# Identification of a RAB32-LRMDA-Commander membrane trafficking complex reveals the molecular mechanism of human oculocutaneous albinism type 7

Rebeka Butkovič [1] ✉, Michael D. Healy [2], Cecilia de Heus[3], Alexander P. Walker[1], Wyatt Beyers [4], Kerrie E. McNally [5], Philip A. Lewis [6], Kate J. Heesom [6], Nalan Liv [3], Judith Klumperman [3], Santiago Di Pietro [4], Brett M. Collins [2] & Peter J. Cullen [1] ✉

The endosomal Commander assembly associates with the sorting nexin-17 (SNX17) cargo adaptor to regulate cell surface recycling of internalised integral proteins including integrins and lipoprotein receptors. Here, we identify leucine rich melanocyte differentiation associated (LRMDA) as a Commander binding protein. We reveal that LRMDA and SNX17 share a common mechanism of Commander association, and that LRMDA simultaneously associates with Commander and active RAB32, establishing distinct RAB32-LRMDA-Commander and SNX17-Commander assemblies. Functional analysis in melanocytes reveals distinct roles for RAB32-LRMDA-Commander and SNX17-Commander in melanosome biogenesis. We reveal how LRMDA mutations, causative for oculocutaneous albinism type 7, a hypopigmentation disorder accompanied by poor visual acuity, uncouple RAB32 and Commander binding thereby establishing the mechanistic basis of this disease. Our discovery of this alternative Commander assembly highlights the plasticity of Commander function in human pigmentation and extends the Commander function beyond the SNX17-mediated regulation of cell surface proteome.

Commander is a super-complex composed of sixteen subunits that mediates the endosomal recycling of diverse transmembrane cargo proteins, including integrins and lipoprotein receptors, from endosomes to the plasma membrane[1–3]. It is assembled from the Retriever heterotrimer (VPS35L, VPS26C and VPS29)[4], the dodecameric CCC complex (CCDC22, CCDC93, COMMD1-10)[5], and DENND10[6–9]. In humans, mutations in genes encoding Commander are causative for Ritscher-Schinzel syndrome (RSS), a complex developmental disease associated with a core triad of cerebellar-cardiac-craniofacial malformation (hence it is alternatively named 3C-syndrome)[10–12]. Causative mutations reduce the stability of Retriever and the CCC complex and perturb Commander assembly, leading to decreased efficiency of cargo recycling and consequently reduced cell surface presentation of key functional proteins[6,13].

[1]School of Biochemistry, Faculty of Life Sciences, Biomedical Sciences Building, University of Bristol, Bristol BS8 1TD, UK. [2]Centre for Cell Biology of Chronic Disease, Institute for Molecular Biosciences, The University of Queensland, St. Lucia, QLD 4072, Australia. [3]Center for Molecular Medicine, University Medical Center Utrecht, Utrecht University, 3584 CX Utrecht, The Netherlands. [4]Department of Biochemistry and Molecular Biology, Colorado State University, 111 MRC Building, 1870 Campus Delivery, Fort Collins, Colorado 80523-1870, USA. [5]MRC Laboratory of Molecular Biology, Cambridge, UK. [6]Bristol Proteomics Facility, School of Biochemistry, Faculty of Life Sciences, Biomedical Sciences Building, University of Bristol, Bristol BS8 1TD, UK. ✉e-mail: rebeka.butkovic@bristol.ac.uk; pete.cullen@bristol.ac.uk

Cargo recognition within the Commander endosomal recycling pathway is principally mediated through the cargo adaptor sorting nexin-17 (SNX17)[4]. SNX17 recognises ØxNxx[YF] sorting motifs in the cytoplasmic domains of transmembrane proteins as they arrive in the limiting membrane of the endosome vacuole (where Ø = hydrophobic residue; x = any residue)[1,4,14,15]. We, and others, recently described how SNX17 directly associates with the Retriever subcomplex of Commander to facilitate transmembrane cargoes recycling to the plasma membrane[1,2,16]. Structurally, this is achieved through the binding of a carboxy-terminal $^{465}$IGDEDL$^{470}$ sequence in SNX17 (where Leu470 is the terminal carboxy residue) into a conserved pocket in the VPS35L subunit of Retriever composed of Arg248, Trp280 and Lys283[1,2,4,16].

In contrast to the numerous accessory proteins that associate with Retromer, a structural homologue of Retriever, only two adaptors, the ubiquitously expressed SNX17 and the more tissue-specific SNX31, have been identified, of which only SNX17 has, to date, been characterised for the Commander trafficking pathway. Here, we identify a distinct Commander complex defined by Leucine-rich melanocyte differentiation-associated protein, LRMDA (*aka* C10orf11), binding to VPS35L and thereby the entire Commander assembly. We define the mechanism by which LRMDA associates with Commander and establish the mutually exclusive nature of SNX17-Commander and LRMDA-Commander assemblies, and how LRMDA independently associates with the active form of RAB32[17]. We demonstrate that LRMDA coupling to Commander is essential for the RAB32-dependent biogenesis of melanosomes and show that LRMDA mutations, causative for oculocutaneous albinism type 7 (OCA7), a hypopigmentation disorder accompanied by poor visual acuity[18], uncouple RAB32 and Commander binding thus establishing the mechanistic basis of this disease. With parallel analysis of SNX17, we reveal distinct SNX17-dependent and LRMDA-dependent roles for Commander during melanosome biogenesis. We propose that LRMDA acts as an adaptor for recruiting the Commander assembly to membranes containing active RAB32. By discovering and functionally validating this alternative RAB32-LRMDA-Commander pathway we provide additional molecular insight into the plasticity of Commander function within a complex organelle biogenesis pathway, dysregulation of Commander in human disease, and identify connectivity between this endosomal sorting complex and the RAB32-family of regulatory GTPases. Our work opens unexplored avenues for mechanistic analysis of the functional role of RAB32 in an array of membrane transport pathways.

## Results

### In silico screening identifies LRMDA as a Retriever interactor

Our previous analyses revealed that conserved residues at the disordered extreme carboxy terminus of SNX17 are critical for binding to the Retriever sub-complex of Commander. SNX31, a SNX17 homologue with cell type-limited expression was also recruited to Retriever via a highly similar sequence ($^{435}$IKEEDL$^{440}$)[1,4]. To identify proteins with similar sequence motifs we performed an in silico human proteome-wide screen using ScanProsite, searching for an [IL]$^{-5}$-x$^{-4}$-[DE]$^{-3}$-x$^{-2}$-x$^{-1}$-[IL]$^{0}$> motif, where '[IL]>' defines the very carboxy-terminal residues as either leucine or isoleucine and 'x' corresponds to any residue (Fig. 1A). This motif was based on our previous functional and structural analysis which demonstrated that: i) there is a high degree of sequence conservation in this region, ii) residues in the −5, −3 and 0 positions make critical interactions with Retriever that when substituted result in a loss of interaction, and iii) the carboxyl group makes an additional contact that stabilises the interface, suggesting the positioning of the motif at the extreme C-terminus is critical for interaction, reminiscent of PDZ binding motifs[1,4].

The ScanProsite search identified 51 possible interactors, including SNX17 and SNX31 as expected (Fig. 1B). We next used AlphaFold2 to predict which of these identified proteins were able to form an interface with Retriever, modelling the final 30 residues of each hit and

evaluating their ability to bind to a heterodimer of VPS35L:VPS26C by assessing the ipTM scores[19,20]. The majority of predicted assemblies placed the disordered carboxy-termini into the previously defined SNX17-binding pocket at the VPS35L:VPS26C interface[1,2,16], however, only three produced a high-confidence prediction in all 5 models: SNX17, SNX31 and LRMDA (Fig. 1B). For LRMDA, AlphaFold2 modelling predicted that the IRDDQL$^{226}$ motif (where Leu226 is the carboxy-terminal residue) bound to the same site in Retriever as SNX17, with the terminal Leu226 binding into the conserved pocket of VPS35L defined by Arg248, Trp280 and Lys283[1,2,16] (Fig. 1C, Supplementary Fig. 1A).

LRMDA (Leucine Rich Melanocyte Differentiation Associated) consists of an N-terminal leucine-rich-repeat (LRR) domain with an extended C-terminal tail that includes the predicted Retriever-binding motif (IRDDQL$^{226}$). As with SNX17, the C-terminal residue, Leu226, is tightly buried in the VPS35L:VPS26C interface and its carboxyl group makes a direct and necessary contact with Arg248 of VPS35L (Fig. 1C). To experimentally validate the predicted binding of LRMDA to Retriever, we generated GFP-LRMDA and LRMDA-GFP fusion constructs and used them in GFP-nanotrap immunoprecipitation assays. Similarly to GFP-SNX17[1,4] amino-terminally tagged GFP-LRMDA pulled down VPS35L and VPS26C, however, the carboxy-terminally GFP-tagged LRMDA, LRMDA-GFP, failed to associate with these Retriever subunits (Fig. 1D). The critical importance of the carboxy-terminal leucine residue was further established through mutagenesis; like GFP-SNX17(L470G)[1,4], a GFP-LRMDA(L226G) mutant displayed a substantial loss of Retriever binding (Fig. 1D).

Finally, we quantified the interactome of GFP-LRMDA in RPE1 cells using GFP-nanotrap immuno-isolation and unbiased TMT-proteomics (Supplementary Data 1). This revealed that GFP-LRMDA was able to pull down Retriever and all remaining subunits of the Commander super-assembly (Fig. 1E). Together these data identify that LRMDA coupling to Retriever is sufficient to engage the entire Commander super-assembly through a mechanism shared with SNX17. Due to this shared mechanism of Retriever association, we speculate that Retriever cannot interact with both LRMDA and SNX17 simultaneously, thus delineating two distinct Commander assemblies: the canonical SNX17-Commander and the non-canonical LRMDA-Commander.

### LRMDA links Retriever to RAB32

In addition to identifying Commander, our proteomic quantification of the LRMDA interactome revealed another highly enriched binding partner, RAB32 (Fig. 1E, Supplementary Data 1). This small GTPase has previously been shown to interact with LRMDA during melanosome maturation[17]. However, RAB32 has not previously been linked to Commander, or shown to have a functional role in Commander-mediated endosomal sorting.

To identify if RAB32 could simultaneously assemble with LRMDA and Retriever, we modelled full-length LRMDA, Retriever and RAB32 using AlphaFold2. A high-confidence model predicted that LRMDA assembled with RAB32 through the LRR domain at the amino-terminus of LRMDA, while its disordered carboxy-terminal tail (residues 200-205) made a β-sheet extension with VPS26C to position the carboxy-terminal Leu226 into the conserved pocket in VPS35L (Fig. 2A, Supplementary Fig. 5A). Notably, the same β-sheet extension has not been observed for the C-terminal residues of SNX17[1,2,16]. To probe this model, we truncated the previously described GFP-LRMDA and LRMDA-GFP constructs generating two LRMDA truncations where we deleted the disordered tail region, GFP-LRMDA$^{1-176}$, or deleted the amino-terminal LRR-domains, GFP-LRMDA$^{127-226}$ (Fig. 2B). Following GFP-nanotrap immunoprecipitation both GFP-LRMDA and LRMDA-GFP displayed association with RAB32 even though LRMDA-GFP failed to associate with Retriever (Fig. 2C). Consistent with the AlphaFold2 prediction, the tail-deletion mutant GFP-LRMDA$^{1-176}$ retained RAB32 binding, but not Retriever and conversely, the LRR-deletion mutant

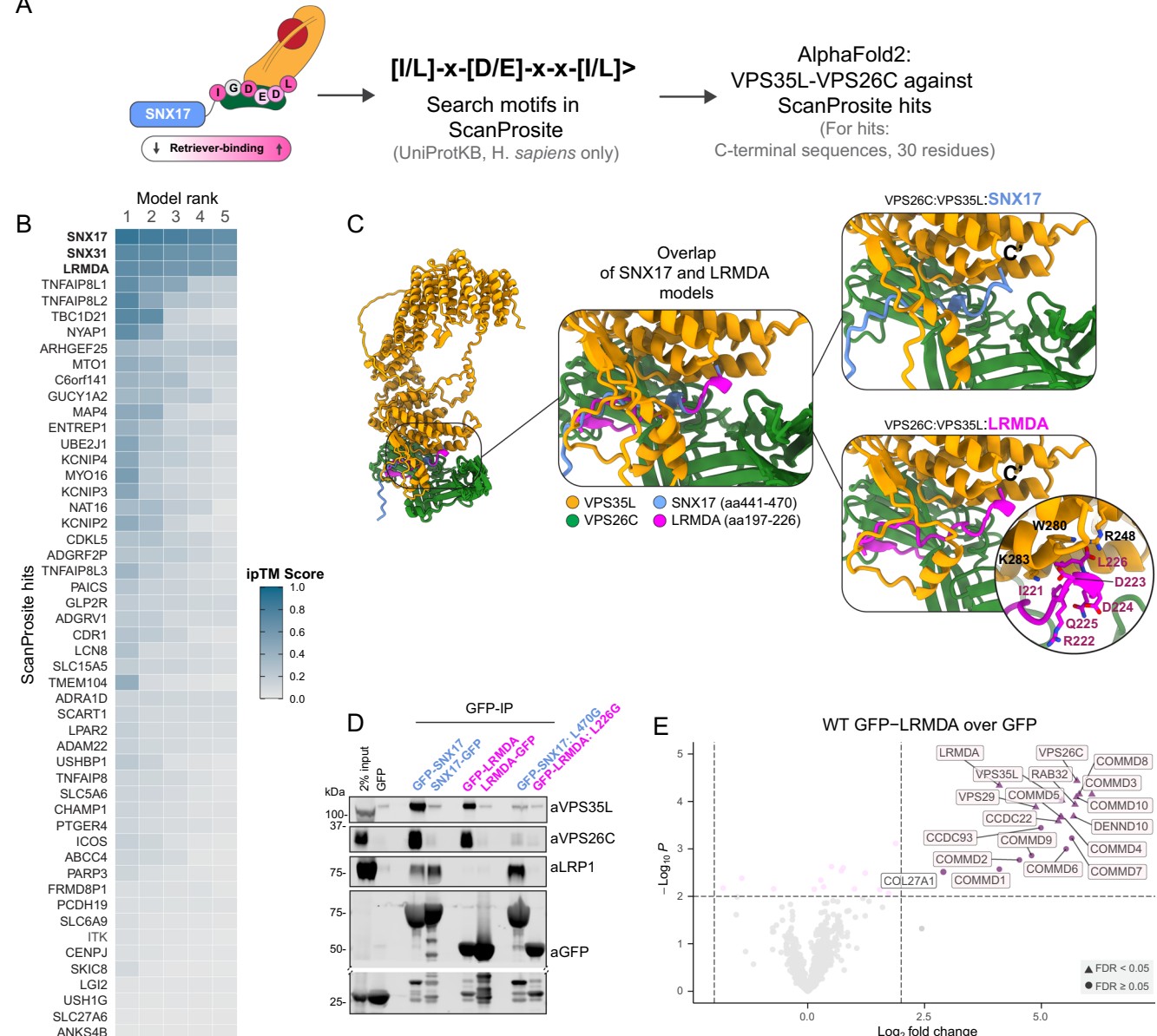

**Fig. 1 | Bioinformatic search identifies LRMDA as a putative Retriever binding partner. A** Schematic depicting the screening process. The C-terminal sequence of SNX17 was used as a template to search for other proteins with Retriever-binding motif, and the C-terminal sequences of hits identified through ScanProsite search were analyzed for VPS35L-VPS26C binding using AlphaFold2 multimer. **B** HeatMap summarising results of ScanProsite search and AlphaFold analysis. The 'per-chain' ipTM scores are given for each ScanProsite hit, for each of the five ranks. Higher ipTM scores suggest higher confidence of modelling. **C** Overlap of two AlphaFold2 models, comparing the interaction between LRMDA-VPS35L:VPS26C and SNX17-

VPS35L:VPS26C. **D** HEK293T cells were transfected with N- or C-terminally GFP-tagged LRMDA or SNX17 and mutants GFP-SNX17 L470G or GFP-LRMDA L226G. The lysates were used in GFP-trap experiments to analyze the association with the Retriever complex, and SNX17 cargo protein LRP1. Data representative of 2 independent repeats. **E** Volcano plot showing the enrichment of proteins in GFP-LRMDA pulldown in RPE1 cells, compared to the GFP sample. $n = 4$, two-tailed paired t-test, significantly enriched proteins with $p < 0.01$ and Log2 fold change > 2 are labelled. Proteins with FDR < 0.05 are depicted with triangles.

GFP-LRMDA[127-226] lost RAB32 binding but retained Retriever association (Fig. 2C).

The initial modelling of the last 30 residues of the LRMDA carboxy-terminus with Retriever predicted engagement of the carboxy-terminal IRDDQL[226] motif to the VPS35L:VPS26C interface at the amino-terminal end of the extended VPS35L α-solenoid (Fig. 1C). The modelling of full-length LRMDA revealed additional contacts between VPS26C and the disordered LRMDA sequences that were not included in the initial modelling (Fig. 2A, Supplementary Fig. 1A). To extend our validation of the LRMDA and Retriever interface presented in Figs. 1D and 2C, we generated additional site-directed mutants in the conserved LRMDA tail, GFP-LRMDA I221A, D223A, and L189D. Whilst

Ile221 and Asp223 are predicted to bind at the interface between VPS26C and VPS35L or the VPS35L pocket itself, Leu189 is predicted to interact with VPS26C at the side facing away from its interface with VPS35L (Fig. 2A, Supplementary Fig. 1A). In agreement with the AlphaFold2 prediction, all mutations displayed a loss in Retriever association but retained binding to RAB32 (Fig. 2D). To test the predicted AlphaFold2 model from the view of Retriever, we used previously generated mutations within VPS35L that perturb the SNX17 association[1]. Consistent with SNX17 and LRMDA engaging the same pocket within VPS35L, all three point mutants, VPS35L-GFP R248A, W280A and K283E displayed a significant decrease in the pull-down of overexpressed mCherry-LRMDA (Fig. 2E).

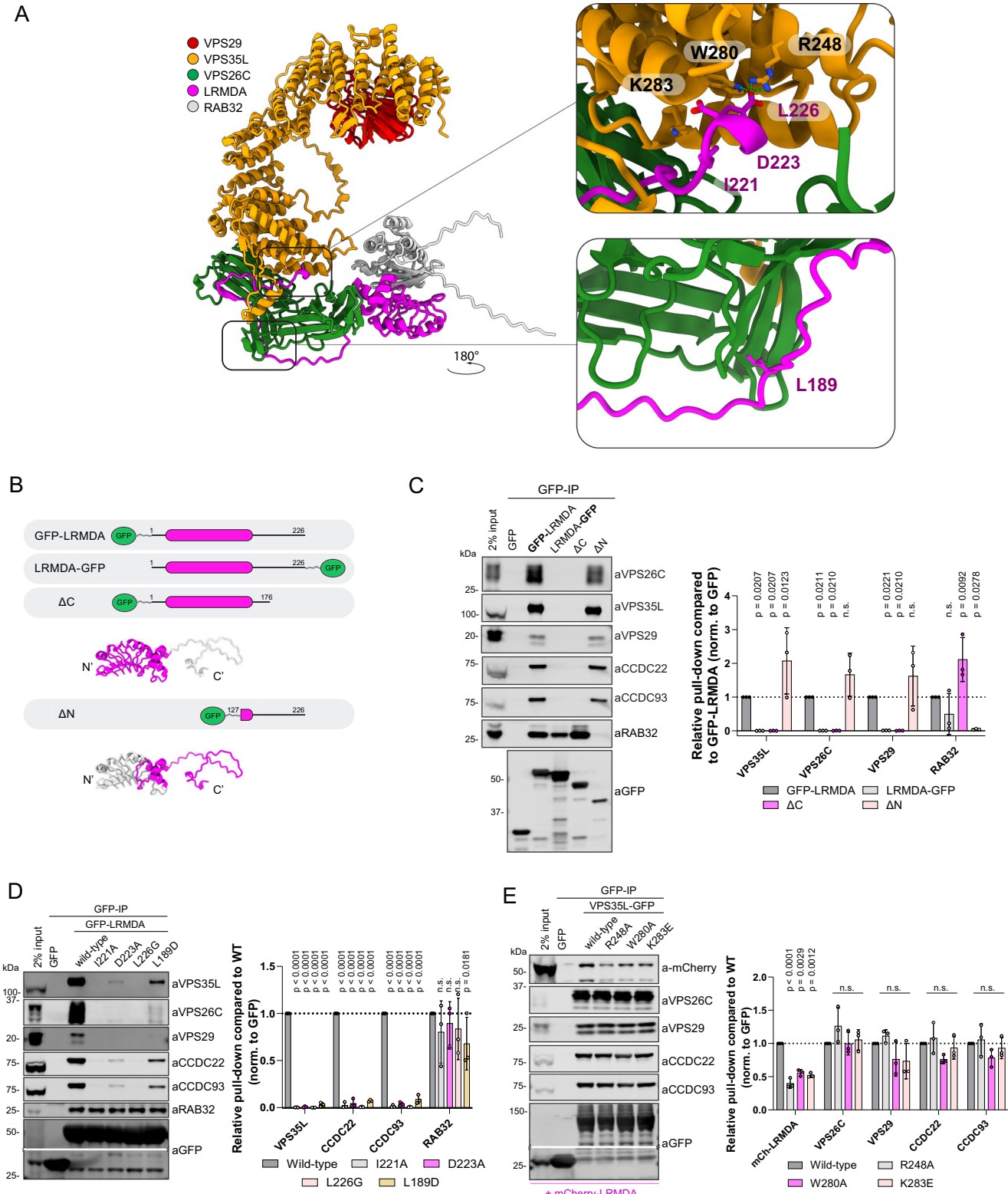

**Fig. 2 | LRMDA employs a SNX17-like motif to associate with the Retriever complex. A** AlphaFold2 model showing the assembly of Retriever (VPS35L, VPS26C and VPS29) with LRMDA and RAB32. **B** Schematic depicting constructs used in panel **C**. **C** HEK293T cells were transfected with N- or C-terminally GFP-tagged LRMDA, or truncated forms of GFP-LRMDA, and the lysates were used in GFP-trap experiments to analyze association with Retriever complex or RAB32. $n = 3$ biological replicates, 2-way ANOVA with Dunnett's multiple comparison test, data presented as mean values and error bars represent s.d., non-significant, or n.s., denotes changes with $p > 0.05$. **D** HEK293T cells were transfected with GFP, GFP-LRMDA or GFP-LRMDA I221A, D223A, L226G or L189D. The lysates were used in

GFP-trap experiments to analyze the association with the Retriever complex or RAB32. $n = 3$ biological replicates, 2-way ANOVA with Dunnett's multiple comparison test, data presented as mean values and error bars represent s.d.; n.s. - changes with $p > 0.05$. **E** HEK293T cells were co-transfected with mCherry-LRMDA and GFP, wild-type VPS35L-GFP or mutant VPS35L-GFP R248A, W280A or K283E. The lysates were used in GFP-trap experiments to analyze association with mCherry-LRMDA or Retriever complex. $n = 3$ biological replicates, 2-way ANOVA with Dunnett's multiple comparison test, data presented as mean values and error bars represent s.d.; n.s. - changes with $p > 0.05$.

Next, we sought to validate the predicted interaction between LRMDA and RAB32. GFP-LRMDA point mutants Y106D, L119D and F145D (Fig. 3A, Supplementary Fig. 5B) all lost the ability to associate with RAB32, while retaining binding to Retriever (Fig. 3B). Likewise, RAB32 mutations targeting the predicted LRMDA interface, GFP-RAB32 D61L, F62D and L64D, displayed significantly perturbed association with mCherry-LRMDA and were unable to incorporate into the LRMDA-Retriever assembly as exemplified by the loss of VPS35L association (Fig. 3C).

RAB32 is a small GTPase that associates with its effectors through a GTP-dependant mechanism; GTP binding results in a conformational change that stabilises the switch I and switch II regions[21,22]. Thus, we investigated whether the RAB32 activity state influenced the association with LRMDA. Indeed, in comparing our model with previously experimentally resolved active and inactive conformations of RAB32, we found that AlphaFold2 modelling was broadly consistent with the active, GTP-bound, conformation of RAB32 binding to LRMDA (Supplementary Fig. 1B). Furthermore, consistent with RAB32 binding to its other effector proteins, such as LRRK2 and VARP[23,24], the binding of LRMDA is predicted to engage switch I, switch II and the inter-switch regions. Point mutations in RAB32, D61L, F62D and L64D, residing in the switch I/Inter-switch region that undergoes localised structural rearrangement upon GTP binding (Supplementary Fig. 1C, D), confirmed their importance for LRMDA interaction (Fig. 3C). To test the effect of RAB32 activation on LRMDA binding, we used dominant inactive and active RAB32 mutants, T39N and Q85L respectively (Fig. 3A, Supplementary Fig. 1C). The active GFP-RAB32 Q85L mutant bound to mCherry-LRMDA, whilst the inactive mutant lost association with LRMDA and Retriever (Fig. 3D). Whilst we did not detect the RAB32-related RAB38 in our LRMDA interactome performed in RPE1 cells (Fig. 1E, Supplementary Data 1), where the expression of RAB38 is low, the interfacial residues in RAB32 are conserved in RAB38 (Supplementary Fig. 1E). Moreover, the AlphaFold2 modelling supports the formation of a RAB38-LRMDA heterodimer and predicts a similar engagement of residues RAB38 Asp45, Phe46 and Leu48 (analogous to RAB32 Asp61, Phe62 and Leu64) at the interface with LRMDA, which maintain similar orientations and conformation in both GTP-RAB32 and GTP-RAB38 (Supplementary Figs. 1F, G, 4C). Indeed, we could detect moderate enrichment of RAB38 in the GFP-LRMDA pull-down, performed in MNT1 cells, where RAB38 is expressed at higher levels (Supplementary Fig. 1H). Moreover, GFP-LRMDA F145D failed to immunoprecipitate RAB38 (Supplementary Fig. 1H). These findings are consistent with previous analysis of LRMDA binding to active RAB32 and RAB38, with a possibly weaker affinity for LRMDA associating with RAB38[17].

Finally, to establish the direct nature of the binding between LRMDA, Retriever and RAB32, we utilised insect cell expression to purify full-length recombinant LRMDA and RAB32 (Supplementary Fig. 2A, B) and, using our published methodology[6], the heterotrimeric Retriever complex incorporating His-tagged VPS29. Using TALON resin to immobilise Retriever, we added purified LRMDA and/or RAB32 (Fig. 3E, Supplementary Fig. 2C). The resulting pull downs established that LRMDA directly bound to Retriever. Conversely, RAB32 failed to associate with Retriever in the absence of LRMDA but was enriched in the pull-down when LRMDA was included. Notably, under these experimental conditions, LRMDA binding to Retriever was slightly increased in the presence of RAB32 (Fig. 3E). Overall, these data establish that LRMDA acts as a direct linker between RAB32 and Retriever and reveal the molecular mechanism for the assembly of this RAB32:LRMDA:Retriever complex.

To evaluate whether LRMDA can act as a linker to aid the recruitment of Retriever onto RAB32-positive membranes, we utilised live imaging to investigate the localisation of GFP-RAB32, mCherry-LRMDA, and VPS35L-GFP in melanocytes, where RAB32 and LRMDA function in the early stages of melanosome biogenesis[17]. By co-expressing GFP-RAB32 with wild-type mCherry-LRMDA or mCherry-LRMDA L226G in MNT1 LRMDA KO cells, we observed that both wild-type and mutant LRMDA were able to associate with RAB32-positive melanosomes, consistent with previous analysis (Supplementary Fig. 3A)[17]. Next, we co-expressed VPS35L-GFP and mCherry-LRMDA in MNT1 LRMDA knock-out cells and observed partial colocalisation between wild-type VPS35L and LRMDA (Supplementary Fig. 3B), with VPS35L-GFP present in MNT1 cells in two populations: as bright puncta in proximity to melanosomes and partially colocalising with mCherry-LRMDA, and as a more diffuse signal co-localising with mCherry-LRMDA. In contrast, mCherry-LRMDA L226G expressed in MNT1 LRMDA KO cells exhibited significantly decreased colocalisation with VPS35L-GFP (Supplementary Fig. 3B).

## Cooperativity in binding within the RAB32:LRMDA:Retriever complex

To test the selectivity of key site-directed mutants we used unbiased proteomics to quantify the interactomes of GFP-LRMDA F145D and GFP-LRMDA L226G. GFP-LRMDA F145D lost only the ability to bind RAB32, establishing the highly selective nature of this association, whilst GFP-LRMDA L226G displayed a significant decrease in the binding of all 16 Commander subunits and a less prominent, but significant, decrease in RAB32 association (Fig. 4A, B, Supplementary Data 1). To further investigate the influence of LRMDA binding to Retriever on its interaction with RAB32, we next expressed wild-type GFP-LRMDA, or the two truncated mutants GFP-LRMDA[1-176] (ΔC) and GFP-LRMDA[127-226] (ΔN) (Fig. 2B) in wild-type HeLa cells or CRISPR-Cas9 VPS26C knock-out HeLa cells (Supplementary Fig. 3C). As shown by pull-downs in parental HeLa cells, GFP-LRMDA pulled down several Commander components. Conversely, in the VPS26C KO cells the interaction with Commander could not be detected, reflecting the importance of the VPS26C-VPS35L interface for the LRMDA association. In the VPS26C KO cells, wild-type GFP-LRMDA also failed to associate with RAB32, but the LRMDA[1-176] truncation bound to RAB32. Expectedly, the LRMDA[127-226] truncation could not bind to Commander in VPS26C knock-out cells and could not interact with RAB32. These data, along with the results of our in vitro pull-down assay showing improved LRMDA binding in the presence of RAB32 (Fig. 3E), reveal that the association between Retriever and LRMDA promotes binding to RAB32, and conversely, RAB32 binding enhances LRMDA association with Retriever. While we have no mechanistic explanation for these observations, they provide evidence for a level of cooperativity in the binding between LRMDA, RAB32 and Retriever, likely ensuring the formation of stable and functional RAB32:LRMDA:Retriever complex

## OCA7 causative mutations perturb the assembly of the RAB32:LRMDA:Retriever complex

Mutations in LRMDA lead to a rare congenital human developmental disorder, oculo-cutaneous albinism VII (OCA7). We therefore explored where the OCA7-causative mutations map on the LRMDA structure and the consequences of these mutations on LRMDA interactions with Retriever and RAB32. Several reported patient mutations in LRMDA are early truncations or lead to frameshifts (e.g. A51fs, R86*) likely affecting the folding and stability of LRMDA (Fig. 4C and D)[18,25,26]. Hence, in our analysis, we only included single-residue substitutions and truncations that occur within the disordered carboxy-terminal region. One of the mutations, LRMDA N117K, is predicted to lie at the interface with RAB32, whereas Y208*, G217S and R222* cluster in the unstructured carboxy-terminal region (Fig. 4D). In GFP-nanotrap analysis of OCA7-causative mutants, LRMDA N117K displayed a loss of detectable RAB32 binding but retained association with Retriever, consistent with structure predictions (Fig. 4E). In contrast, the carboxy-terminal mutants all lost association with Retriever and the CCC complex (i.e.

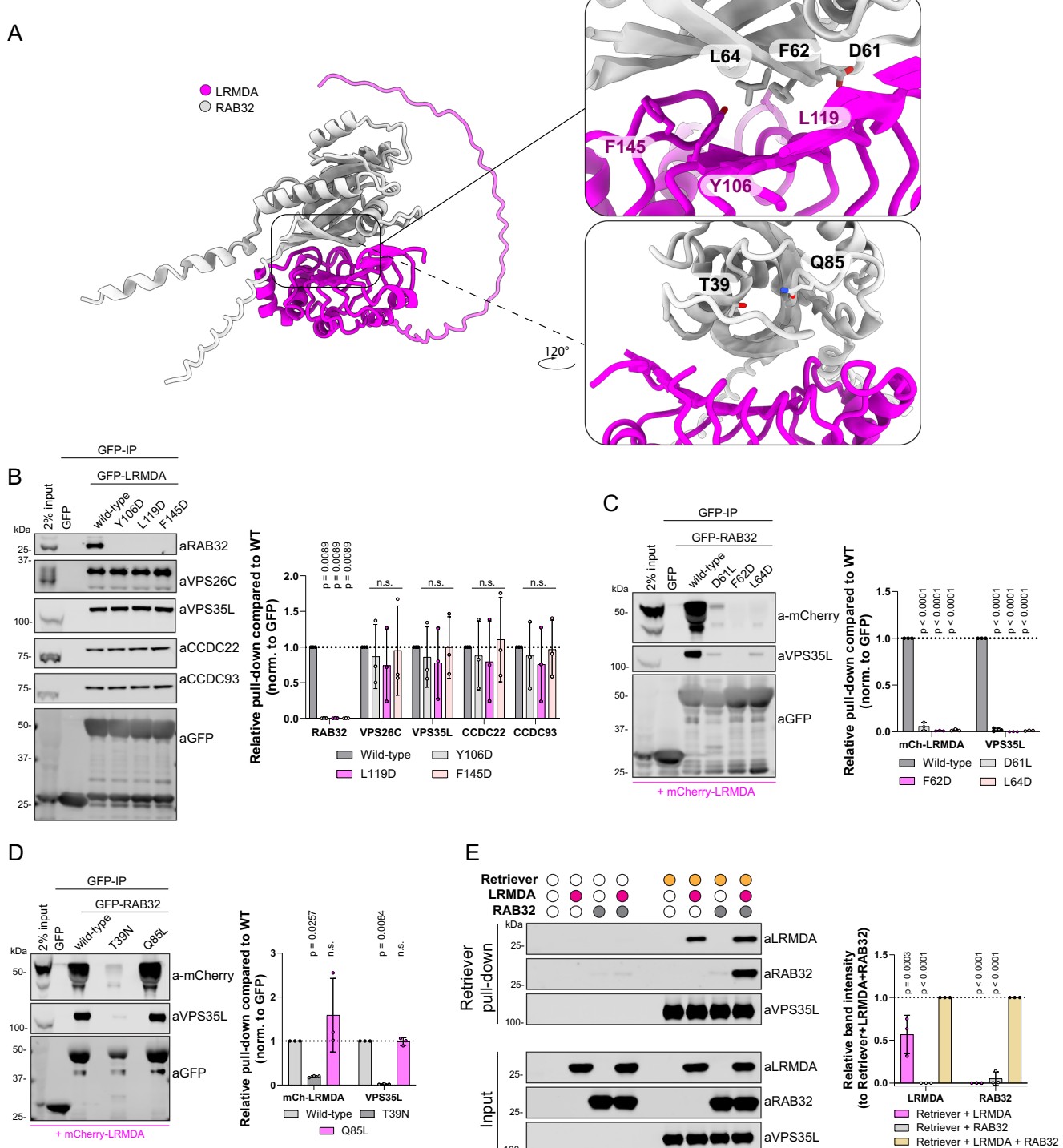

**Fig. 3 | LRMDA interacts with RAB32 through N-terminal leucine rich repeats and acts as a linker between RAB32 and Retriever. A** AlphaFold2 model showing the assembly of LRMDA and RAB32, with top panel highlighting the interfacial residues (**B**, **C**) and bottom panel showing the position of RAB32 T39 and Q85 residues that were substituted to generate constitutively inactive and active RAB32 mutants, respectively (**D**). **B** HEK293T cells were transfected with GFP, GFP-LRMDA or mutants of the interfacial residues to probe RAB32 interaction: GFP-LRMDA Y106D, L119D, F145D. The lysates were used in GFP-trap experiments to analyze the association with the Retriever complex or RAB32. $n = 3$ biological replicates, 2-way ANOVA with Dunnett's multiple comparison test, data presented as mean values and error bars represent s.d.; n.s., denotes changes with $p > 0.05$. **C** HEK293T cells were transfected with GFP, GFP-RAB32 or mutants of the interfacial residues to probe LRMDA interaction: GFP-RAB32 D61L, F62D, L64D. The lysates were used in

GFP-trap experiments to analyze the association with Retriever complex or LRMDA. $n = 3$ biological replicates, 2-way ANOVA with Dunnett's multiple comparison test, data presented as mean values and error bars represent s.d. **D** HEK293T cells were transfected with GFP, GFP-RAB32 or constitutively active GFP-RAB32 Q85L or inactive T39N. The lysates were used in GFP-trap experiments to analyze the association with Retriever complex or LRMDA. $n = 3$ biological replicates, 2-way ANOVA with Dunnett's multiple comparison test, data presented as mean values and error bars represent s.d.; n.s. denotes changes with $p > 0.05$. **E** Recombinantly purified Retriever, containing His-VPS29, was bound to TALON resin. The bead-bound Retriever was mixed with recombinantly purified LRMDA, RAB32 or LRMDA and RAB32. The normalised band intensities were compared to samples with added RAB32 and LRMDA. $n = 3$ biological replicates, 2-way ANOVA with Dunnett's multiple comparison test, data presented as mean values and error bars represent s.d.

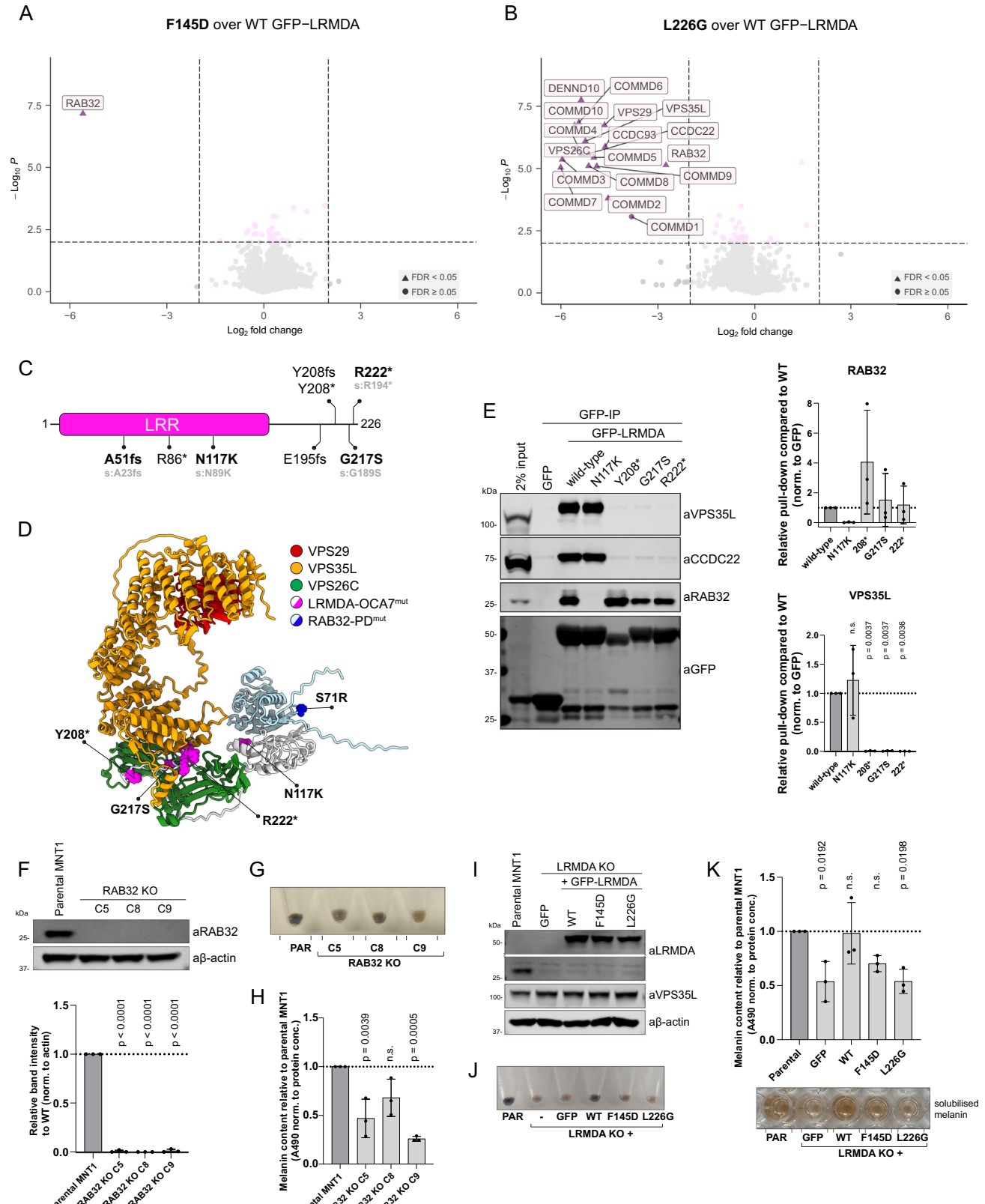

the Commander assembly) but retained their association with RAB32 (Fig. 4E). These results are consistent with the structural analyses, as the truncations Y208*, and R222* remove the necessary C-terminal sequences required for Retriever binding, and the point mutation G217S sits in close association with the VPS26C/VPS35L interface. As an aside, a recent study characterised a mutation within RAB32, S71R, to

be causative for familial Parkinson's disease[27]. This mutation does not appear in the interface with LRMDA and did not interfere with LRMDA-RAB32 interaction (Supplementary Fig. 4B).

Together, these data establish that the OCA7-causative mutations specifically prevent the efficient coupling of RAB32 and Retriever via the LRMDA adaptor. In our proteomic analysis in the human

**Fig. 4 | The direct coupling between RAB32-LRMDA-Retriever is important for human pigmentation. A, B** Volcano plot showing the loss of proteins in GFP-LRMDA F145D (**A**) or GFP-LRMDA L226G (**B**) pulldown in RPE1 cells, compared to wild-type GFP-LRMDA sample. $n = 6$, two-tailed paired t-test, normalised to LRMDA levels. Significantly decreased proteins with $p < 0.01$ and Log2 fold change $< -2$ are labelled. Proteins with FDR $< 0.05$ are depicted with triangles. **C** Schematic depicting the position of reported OCA7-causative mutations in LRMDA. The numbering for the longer LRMDA isoform (Uniprot: A0A087WWI0) is listed in black font, and the shorter isoform (UniProt: Q9H2I8) is listed in grey. **D** Mapping of OCA7-causative mutations in LRMDA onto AlphaFold2 model of RAB32-LRMDA-Retriever assembly. **E** HEK293T cells were transfected with GFP, GFP-LRMDA or OCA7-causative mutations: GFP-LRMDA N117K, Y208stop, G217S or R222stop. The lysates were used in GFP-trap experiments to analyze the association with the Retriever complex or RAB32. $n = 3$ biological replicates, 1-way ANOVA with Dunnett's multiple comparison test, data presented as mean values and error bars represent s.d. **F** RAB32 KO in MNT1 cells validated using western blotting. $n = 3$

biological replicates, 1-way ANOVA with Dunnett's multiple comparison test, data presented as mean values and error bars represent s.d. **G** Cell pellets of different RAB32 KO clones, showing a minor decrease in pigmentation. **H** Measurement of RAB32 KO melanin levels compared to parental MNT1 cell line. $n = 3$ biological replicates, 1-way ANOVA with Dunnett's multiple comparison test, data presented as mean values and error bars represent s.d.; n.s. denotes changes with $p > 0.05$. **I** LRMDA KO MNT1 cells were lentivirally transduced with GFP, wild-type GFP-LRMDA or mutants GFP-LRMDA F145D and L226G, that perturb binding with RAB32 or Retriever, respectively. Data representative of 2 independent repeats. **J** Cell pellets of LRMDA KO cells and cells lentivirally rescued with GFP, wild-type GFP-LRMDA or mutants GFP-LRMDA F145D and L226G, showing a difference in pigmentation. **K** The pellets from (**J**) were used for melanin content measurements. $n = 3$ biological replicates, 1-way ANOVA with Dunnett's multiple comparison test, data presented as mean values and error bars represent s.d.; n.s. denotes changes with $p > 0.05$.

melanogenic cell line MNT1, the quantified GFP-LRMDA interactome only identified the Commander complex components and RAB32, other proteins were absent (Supplementary Fig. 4A, Supplementary Data 1). Assuming the loss-of-function nature of the OCA7-causative mutations, we conclude that the perturbed formation of the RAB32:LRMDA:Retriever complex and association with the entire Commander assembly underlies the pathology observed in OCA7.

## RAB32 and LRMDA facilitate the role of Commander in human pigmentation

Due to the clinical presentation of OCA7 and the main reported functions of both RAB32 and LRMDA related to melanin synthesis and melanosome biogenesis[17,28,29], we explored the functional importance of RAB32:LRMDA:Retriever association for human pigmentation. In agreement with previous studies on RAB32 and LRMDA, we observed a decrease in pigmentation in RAB32 CRISPR/Cas9 knock-out human melanoma MNT1 cells (Fig. 4F–H), as well as in a previously characterised LRMDA knock-out MNT1 cell line (Fig. 4I, J)[17]. We then used the LRMDA knock-out cells and attempted to rescue the decrease in pigmentation observed in these cells by re-expression of wild-type GFP-LRMDA or mutations perturbing the interfaces with either RAB32 or Retriever, F145D and L226G, respectively (Fig. 4I). The melanin content measurements revealed that overexpression of wild-type GFP-LRMDA reinstated melanin synthesis to levels comparable to parental MNT1 cells (Fig. 4J, K). On the other hand, neither LRMDA mutant was able to rescue the pigmentation defect, as both rescued cell lines exhibited visibly reduced melanin levels; however, only the reduction in GFP-LRMDA(L226G) cells was statistically significant, while the decrease in GFP-LRMDA(F145D) cells did not reach significance. Thus, LRMDA-mediated regulation of human pigmentation requires both its direct association with RAB32 and Retriever. We, therefore, propose a model where LRMDA acts as a linker to functionally couple Retriever and the Commander assembly to RAB32-decorated membranes during melanosome biogenesis.

## Retriever function is required for melanosome biogenesis

Melanosomes mature from early endosomes in a process that involves highly regulated trafficking of melanogenic enzymes and other melanosome components, formation of PMEL fibrils and the synthesis of melanin that is supported by de-acidification of late-stage melanosomes[30,31]. Melanosome maturation can be divided into four stages that have been resolved by electron microscopy. Vacuolar domains of early endosomes represent stage I melanosomes and contain the amyloid fibril-forming protein PMEL (premelanosome protein). The PMEL fibrils, which begin to form in stage I melanosomes following a series of proteolytic processing steps, are fully formed and morphologically apparent in stage II melanosomes. Tubular domains of early/recycling endosomes are responsible for the trafficking of

cargo such as the melanogenic enzymes tyrosinase and tyrosinase-related protein 1 (TYR, TYRP1) to stage II melanosomes. In stage III, the melanin synthesis begins, and signature melanin-coated fibrils are noticeable. The melanin deposition onto PMEL fibrils continues in stage IV melanosomes, where fibrils are fully covered by melanin polymers that form a dense melanin core[32]. The mature stage IV melanosomes are then either stored in the cell (retinal pigment epithelial cells) or transferred into the receiving keratinocyte, where melanin acts to protect the nucleus from UV damage (skin melanocytes-keratinocytes)[29,33].

To understand the functional role of Retriever in this complex melanogenic process we generated a VPS35L knock-out model in MNT1 cells, hypothesising that the loss of Retriever would lead to a similar decrease in pigmentation as observed in LRMDA knock-out MNT1 cells (Fig. 5A). In all clonal cell models investigated, VPS35L depletion led to a near total loss of detectable melanin production, a phenotype that was even stronger than in LRMDA knock-out cells (Fig. 5B, C). To investigate this further, we examined parental, LRMDA knock-out, and VPS35L knock-out MNT1 cells using electron microscopy. This revealed an essentially complete loss of late-stage III and IV melanosomes in VPS35L knock-out cells, as characterised by the presence of melanin-decorated PMEL fibrils (Fig. 5D). VPS35L knock-out cells also showed a significant loss of stage II melanosomes. Early endosomes/stage I melanosomes and some abnormal organelles that did not reflect ultrastructural melanosome-staging criteria were the only identifiable organelles. In LRMDA knock-out cells, we noticed overall fewer structures belonging to the melanosome compartment, as expected; however, we could still observe different stages of melanosome maturation, albeit with some abnormalities. The greater, near-complete loss of pigmentation in VPS35L knock-out cells, along with ultrastructural analysis revealing a loss of stage II, III and IV melanosomes, suggested that Retriever has additional roles in melanosome biogenesis and melanin production that are independent of its association with LRMDA and RAB32.

## Commander regulates melanosome biogenesis

To better understand how perturbations in Retriever lead to decreased pigmentation, we performed an interactome analysis of the Retriever complex component GFP-VPS26C in the MNT1 cell line to identify possible melanocyte-specific interactions of Retriever. This analysis, however, did not reveal many additional interactors outside of RAB32 and LRMDA (Fig. 6A, Supplementary Data 1). As LRMDA engages the entire Commander complex, we next generated CCDC22 (a CCC sub-complex subunit) knock-out MNT1 cells (Fig. 6B). Knocking out CCDC22 has previously been shown to reduce the levels of all 16 subunits of Commander. This depletion also resulted in a profound loss of pigmentation, comparable to that observed in VPS35L KO cells (Fig. 6C, D). As Commander-mediated cargo sorting relies on the

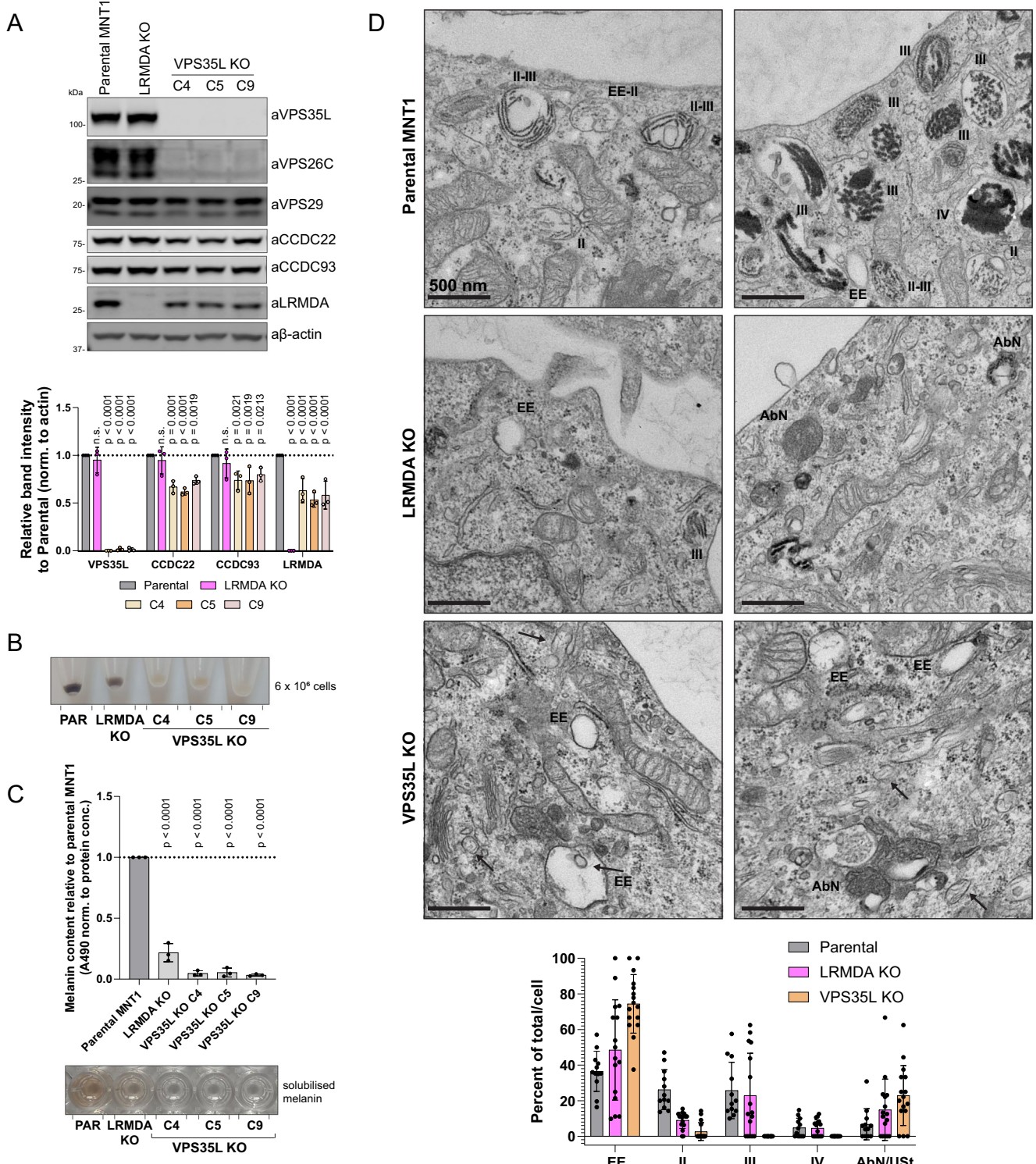

**Fig. 5 | VPS35L KO in melanocytes leads to a significant decrease in pigmentation and perturbs melanosome biogenesis. A** VPS35L KO cells were generated using CRISPR in the MNT1 cell line. The KO was validated using western blotting. *n* = 3 biological replicates, 2-way ANOVA with Dunnett's multiple comparison test, data presented as mean values, and error bars represent s.d.; n.s. denotes changes with *p* > 0.05. **B** Cell pellets of LRMDA KO cells and different VPS35L KO clones show a loss of pigmentation. **C** Cell pellets of LRMDA KO cells and different VPS35L KO clones were used for melanin content measurements. *n* = 3 biological replicates, 1-way ANOVA with Dunnett's multiple comparison test, data presented as mean values and error bars represent s.d. **D** Transmission electron microscopy analysis of parental MNT1 cells and LRMDA KO or VPS35L KO MNT1 cells. Stages of

melanosome biogenesis are marked on EM micrographs: EE – early endosomes/ stage I melanosomes, II – melanosomes with visible PMEL fibrils; III - melanosomes containing melanin-laden PMEL fibrils; IV – melanosomes containing dense melanin core. AbN denotes abnormal melanosome compartments. Arrows point to fibril-like structures in early/endosomes and small vesicles. AbN/USt in the graph denotes either abnormal (AbN) melanosome compartments, or compartments that could not be determined as belonging into one of the four stages – un-staged (USt). Scale bars correspond to 500 nm. *n* = 1 biological replicate (Parental – 12 cells; LRMDA KO – 16 cells; VPS35L KO – 16 cells). Points present the data for individual cells and error bars represent s.d.

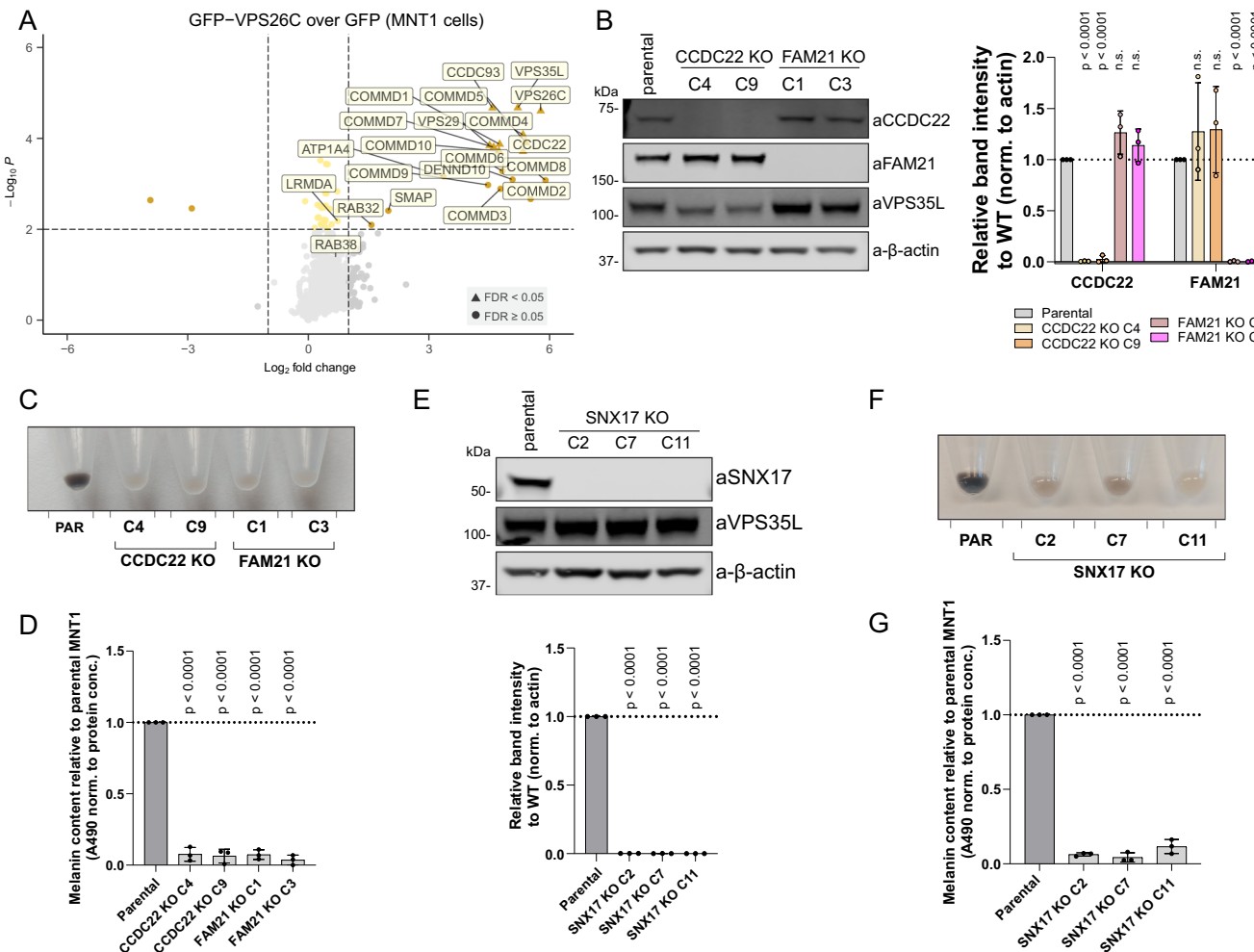

**Fig. 6 | Canonical SNX17-Commander-WASH cargo sorting pathway is important for human pigmentation. A** Volcano plot showing the enrichment of proteins in GFP-VPS26C pulldown in MNT1 cells, compared to the GFP sample. $n = 3$ (biological replicates), two-tailed paired t-test. All significantly enriched proteins with $p < 0.01$ and Log2 fold change > 1 are labelled, along with RAB38 and LRMDA due to their biological relevance. Proteins with FDR < 0.05 are depicted with triangles. **B** CCDC22 KO or FAM21 KO cells were generated using CRISPR in the MNT1 cell line. The KO was validated using western blotting. $n = 3$ biological replicates, 2-way ANOVA with Dunnett's multiple comparison test, data presented as mean values and error bars represent s.d.; n.s. denotes changes with $p > 0.05$. **C** Cell pellets of CCDC22 KO or FAM21 KO clones, show a loss of pigmentation. **D** CCDC22 KO or FAM21 KO cells were used for melanin content measurements. $n = 3$ biological replicates, 1-way ANOVA with Dunnett's multiple comparison test, data presented as mean values and error bars represent s.d. **E** SNX17 KO cells were generated using CRISPR in the MNT1 cell line. The KO was validated using western blotting. $n = 3$ biological replicates, 1-way ANOVA with Dunnett's multiple comparison test, data presented as mean values and error bars represent s.d. **F** Cell pellets of SNX17 KO clones, show a loss of pigmentation, compared to parental MNT1 cells. **G** Cell pellets of SNX17 KO clones were used for melanin content measurements. $n = 3$ biological replicates, 1-way ANOVA with Dunnett's multiple comparison test, data presented as mean values and error bars represent s.d.

interaction with the actin-regulating WASH complex[5,34], we examined the effect of FAM21 (WASH complex component) depletion on pigmentation and observed a significant decrease in melanin content Fig. 6C, D). Due to the more pronounced loss-of-pigmentation phenotype observed in VPS35L, CCDC22 and FAM21 knock-out MNT1 cells, compared to either RAB32 or LRMDA knock-out, and the absence of melanocyte-specific interactors in the Retriever interactome, we explored whether the loss of pigmentation in Commander-deficient cells could be explained by the involvement of the canonical complex of Commander with the SNX17 adaptor in melanosome biogenesis. We generated SNX17 knock-out melanocytes (Fig. 6E), and all three validated clones showed a quantifiable decrease in melanin content (Fig. 6F, G). These data support the model that melanosome biogenesis requires the involvement of both non-canonical RAB32:LRMDA:-Commander, as well as the canonical axis in which the role of Commander depends on SNX17.

To delineate between LRMDA- and SNX17-Commander, we next evaluated the trafficking of the canonical SNX17-Commander cargo, LRP1, which was depleted from the cell surface of SNX17, VPS35L, CCDC22 and FAM21 knock-out MNT1 cells, but not RAB32 or LRMDA knock-out MNT1 cells. This establishes that RAB32-LRMDA-Commander presents a non-canonical assembly of Commander that does not affect trafficking of a classical SNX17-Commander cargo to the cell surface (Supplementary Fig. 4C). Next, we utilised the same cell models to further resolve the perturbations in melanosome biogenesis and investigate the trafficking and processing of the melanosome-specific proteins TYRP1 and PMEL, respectively. The transport of the integral membrane protein TYRP1 is important for melanin synthesis within melanosomes, and previous studies have shown that RAB32, in concert with RAB38, regulates the trafficking of TYRP1 as well as tyrosinase[28]. In parental MNT1 cells, the overlap between the lysosome marker LAMP1 and TYRP1 is typically restricted to a small sub-

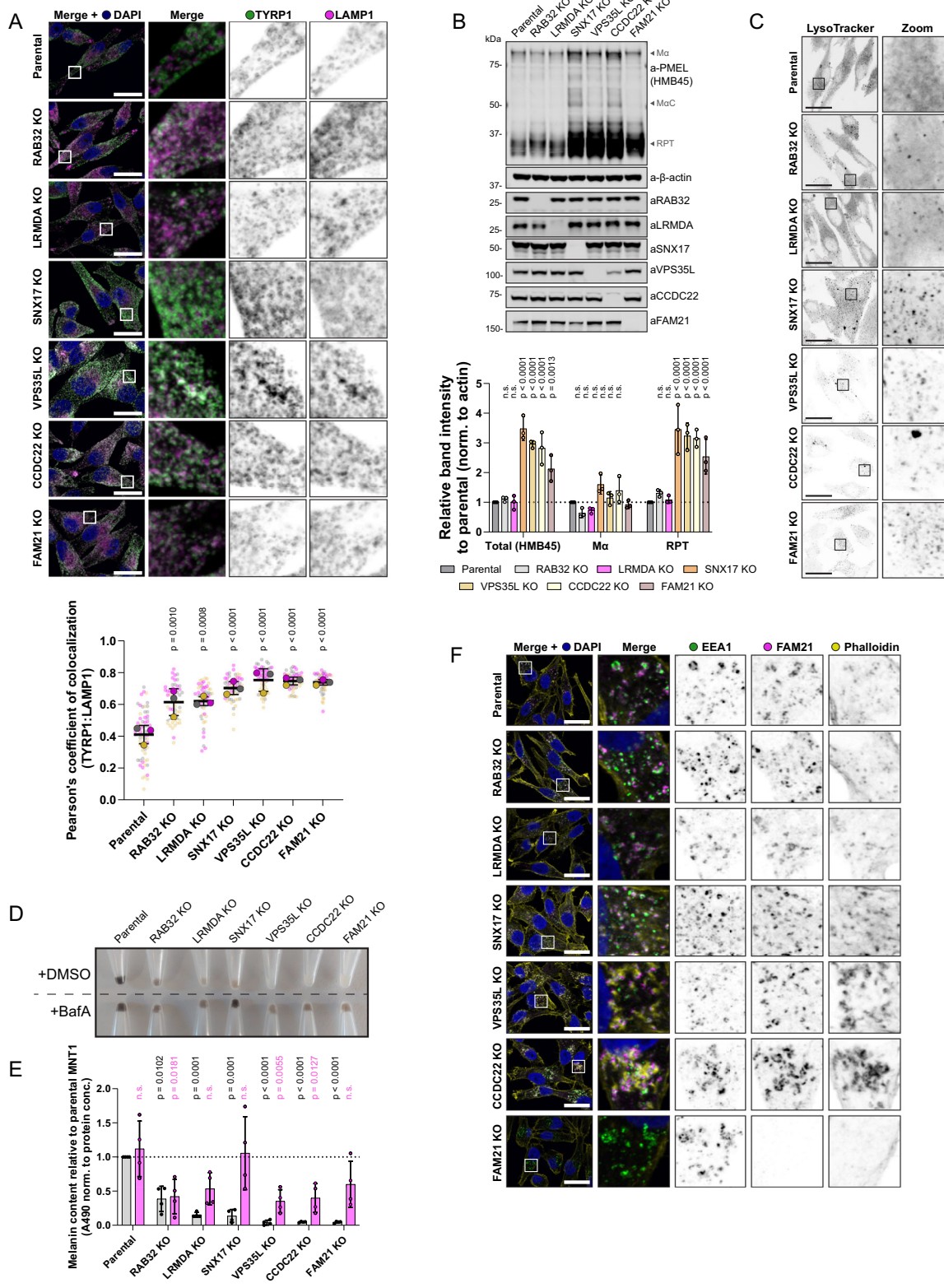

population of organelles, and TYRP1 acts as a marker for late-stage melanosome[30] (Fig. 7A). In comparison with parental cells, we observed an increase in co-localisation of TYRP1 and LAMP1 in all investigated knock-out cell lines, a phenotype that was especially apparent in VPS35, CCDC22 and FAM21 knock-out cells (Fig. 7A). Whilst the perturbed trafficking of TYRP1 was apparent in all knock-out models, we could better de-convolute the phenotypes associated with

either RAB32:LRMDA:Commander or SNX17-Commander by characterising the processing of fibril-forming protein PMEL. We could not observe significant differences in the total amount of PMEL (as detected by the PMEL-HMB45 antibody) in RAB32 or LRMDA knock-out cells, however, we found a significant increase of cleaved PMEL repeat (RPT) fragment levels in SNX17, VPS35L, CCDC22 and FAM21 knock-out cells (Fig. 7B, Supplementary Fig. 4D).

**Fig. 7 | Commander is important for the trafficking of melanogenic enzymes, PMEL processing, compartment neutralisation and dynamic endosomal actin regulation. A** Parental MNT1 cells or RAB32 KO, LRMDA KO, SNX17 KO, VPS35L KO, CCDC22 KO or FAM21 KO cells were immuno-stained for TYRP1 and LAMP1. Scale bar corresponds to 20 μm. *n* = 3 (parental – 67 cells; RAB32 KO – 56 cells; LRMDA KO – 56 cells; SNX17 KO – 56 cells; VPS35L KO – 57 cells; CCDC22 KO – 55 cells, FAM21 KO – 55 cells), large circles represent averages of each independent experiment, and small circles the data for individual cells, colouring corresponds to different independent replicates. 1-way ANOVA with Dunnett's multiple comparison test on data for replicate averages, data presented as mean values and error bars represent s.d. **B** Total PMEL levels and levels of processed PMEL fragments were analyzed in whole-cell lysates from parental MNT1 cells and RAB32 KO, LRMDA KO, SNX17 KO, VPS35L KO, CCDC22 KO or FAM21 KO cells. *n* = 3 biological replicates, 2-way ANOVA with Dunnett's multiple comparison test, data presented as mean values and error bars represent s.d.; n.s. denotes changes with *p* > 0.05.

**C** Parental MNT1 cells or RAB32 KO, LRMDA KO, SNX17 KO, VPS35L KO, CCDC22 KO or FAM21 KO cells were stained with LysoTracker. Fixed cells were observed under a confocal microscope. Representative images are shown; the experiment was performed in three independent replicates. Scale bar corresponds to 20 μm. **D** Parental MNT1 cells or RAB32 KO, LRMDA KO, SNX17 KO, VPS35L KO, CCDC22 KO or FAM21 KO MNT1 cells were treated with DMSO or 100 nM Bafilomycin A1 for 24 h to prevent over-acidification. **E** Cells from (**D**) were used for melanin content measurements. *n* = 4 biological replicates, 2-way ANOVA with Šídák's multiple comparisons test, data presented as mean values and error bars represent s.d.; n.s. denotes changes with *p* > 0.05. **F** Parental MNT1 cells or RAB32 KO, LRMDA KO, SNX17 KO, VPS35L KO, CCDC22 KO or FAM21 KO cells were fixed and immuno-stained for EEA1, FAM21 and F-actin using Phalloidin dye. Representative images are shown; the experiment was performed in three independent replicates. Scale bar corresponds to 20 μm.

In melanocytes, PMEL undergoes a series of proteolytic processing steps, including the cleavage by BACE2 (β-secretase 2) that releases an amyloidogenic Mα fragment into the endosomal lumen (containing N-terminal region (NTR), PKD (polycystic kidney disease domain) and RPT domains), which is further cleaved to produce a fibril-forming CAF (core amyloid fragment) and a fibril-associated RPT fragment[32,35–38]. Whilst several PMEL processing steps are not yet elucidated, many of the identified processing enzymes, such as BACE2, reside in endosomes and require an acidic luminal pH for their activity[30,39,40]. Hence, we speculated that increased processing may be a consequence of over-acidification of an endosome/early-melanosome compartment. Indeed, V-ATPase inhibition through treatment with Bafilomycin A1 (BafA1) prevented the processing of PMEL in parental MNT1 cells, as well as in SNX17 and VPS35L knock-out models (Supplementary Fig. 4E). Furthermore, we used LysoTracker Red DND-99 for the detection of acidified compartments in knock-out MNT1 models. When compared to parental MNT1 cells, the LysoTracker probe revealed a more punctate stain in VPS35L, CCDC22 and FAM21 knock-out cells, and most evidently in SNX17 knock-out cells (Fig. 7C). Whilst we could biochemically observe increased PMEL processing levels, our EM investigation of VPS35L knock-out cells did not reveal an accumulation of fibril-containing melanosomes. However, we did notice short fibril-like structures in early endosomes (stage I melanosomes) and smaller vesicles (arrows in Fig. 5D). The accumulated RPT fragment that we observed does not by itself form amyloids, but rather associates with the fibrils to promote the correct organisation of PMEL sheets onto which the melanin is deposited[41]. It is possible that in SNX17, VPS35L, CCDC22 and FAM21 knock-out cells, PMEL fragments are, like TYRP1, not correctly delivered into the melanosome compartment, or that the low pH environment promotes the PMEL processing, but not the fibril formation.

To investigate the functional importance of over-acidification for melanin production, we again treated the parental and knock-out MNT1 cells with BafA1. The BafA1 treatment rescued the pigmentation decrease observed in SNX17 knock-out cells and offered a partial phenotypic rescue in all knock-out models but not in RAB32 knock-out cells (Fig. 7D, E). In the process of melanosome biogenesis, melanosomes require deacidification to allow for the enzymatic activity of melanogenic enzymes. The ability to rescue the pigmentation decrease observed in SNX17 knock-out cells could indicate that SNX17 is responsible for the trafficking of cargo to the melanosomes that are important for de-acidification or divert cargos that can acidify melanosomes. Additionally, it suggests that SNX17 knock-out is less disruptive to the melanosome biogenesis process than the depletion of Commander components, as in SNX17 knock-out cells all necessary machinery to synthesise melanin appears to be present and can be activated through amelioration of the low pH environment, with BafA1 treatment possibly rescuing de-acidification defect in mature melanosomes, or enhancing tyrosinase activity in immature melanosomes.

## Commander controls melanosome actin dynamics

Finally, as the depletion of Commander components VPS35L or CCDC22 by CRISPR knock-out leads to the loss of pigmentation that can be only moderately rescued by compartment deacidification, we explored the possibility that the Commander complex additionally performs a more global function in the control of melanosome biogenesis. The incomplete rescue of pigmentation defect was also observed in cells lacking FAM21, a WASH complex component which recruits Arp2/3 to endosome to couple Commander-mediated cargo sorting with controlled actin-branching activity[4,5,42,43]. Control of actin dynamics on early/recycling endosomes has previously been reported to be instrumental in the efficient delivery of cargos to later-stage melanosomes and the process of melanosome maturation[44,45]. Moreover, a previous study has identified endosomal actin hyper-accumulation as a feature of Commander knock-out cell models, albeit not in melanocytes[34]. Therefore, to assess the endosomal actin organisation in parental and knock-out MNT1 cell models, we stained the stage I melanosomes/endosomes with EEA1, FAM21 and labelled the filamentous actin with phalloidin. These experiments showed altered endosomal actin in VPS35L knock-out and CCDC22 knock-out cells, with noticeable hyperaccumulation of actin in the proximity of endosomes, but the phenotype was not present in LRMDA knock-out, RAB32 knock-out or SNX17 knock-out cells. In FAM21 knock-out MNT1 cells we observed the expected absence of an endosomal pool of actin (Fig. 7F).

Altogether, we propose that Commander participates in different stages of melanosome biogenesis (Fig. 8): i) efficient melanosome maturation through association with LRMDA and RAB32 and ii) deacidification of the melanosome compartment through a mechanism that requires classic binding to the SNX17 cargo adaptor, and a more global control of melanosome biogenesis through regulation of WASH-mediated actin dynamics.

## Discussion

This study describes the identification of a Commander assembly, defined by the association of LRMDA with Retriever through a mutually exclusive mechanism to the classical binding of Retriever to the endosomal cargo adaptor SNX17[1,2,16]. Through unbiased proteomic analysis, AlphaFold modelling, and in vitro binding analysis, we have revealed that LRMDA directly links Commander and RAB32 and established the molecular mechanism for the interactions between Retriever-LRMDA and RAB32-LRMDA. Through mutant-rescue experiments, we have shown that the function of LRMDA-RAB32 in pigmentation relies on the direct association with the Commander complex, and furthermore, we have established that OCA7-causative mutations in *LRMDA*[18,25] lead to a loss of LRMDA association with RAB32, Retriever and the Commander assembly. Altogether, this provides evidence for Commander acting as an important element within the RAB32-LRMDA-mediated control of human melanosome

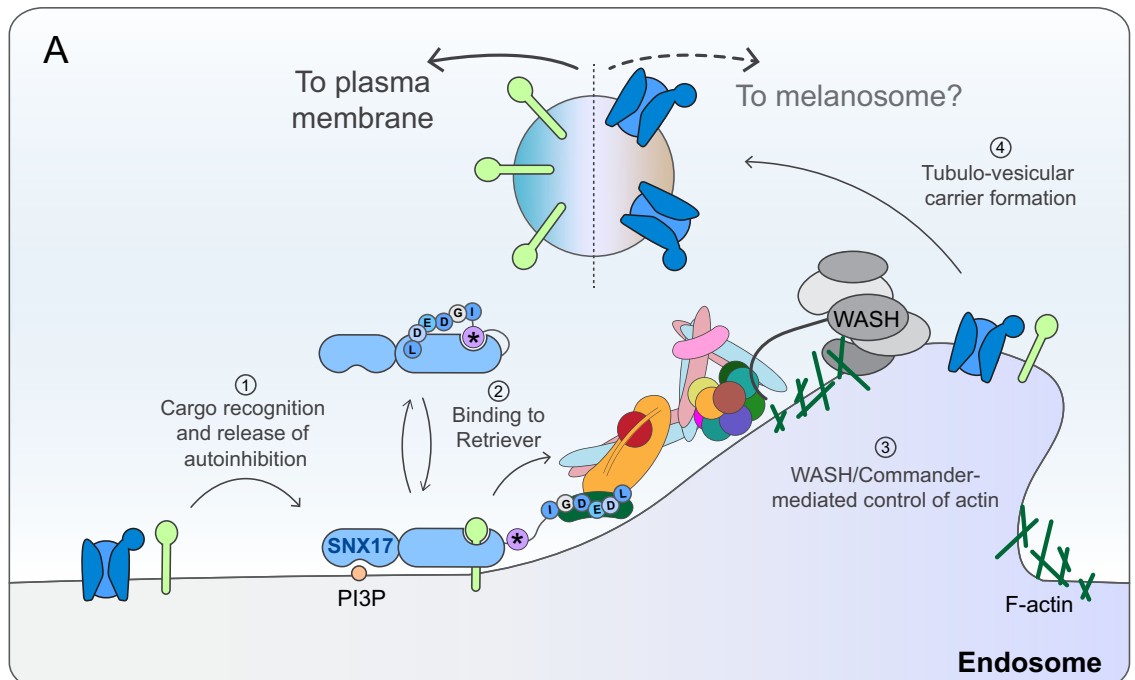

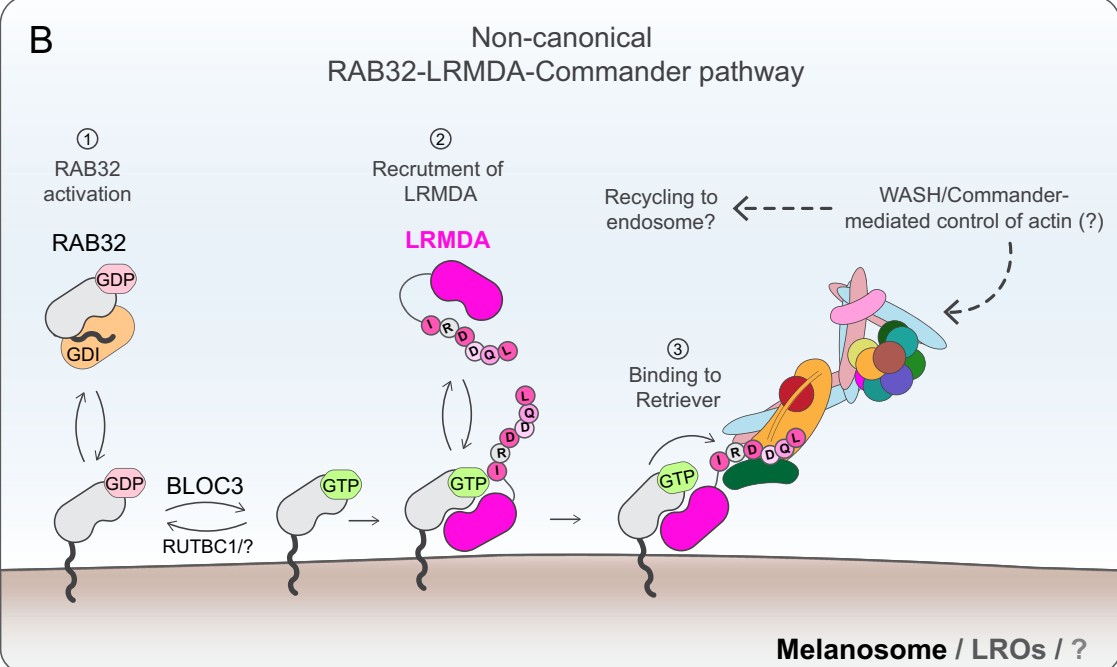

**Fig. 8 | Proposed function for SNX17-Commander and RAB32-LRMDA-Commander in melanosome biology. A** The Commander complex associates with the cargo adaptor SNX17 to mediate the retrieval of cargo from early endosomes to the plasma membrane[1,2,4,16]. In this process, SNX17 associates with the endosomal membrane through recognition of phosphatidylinositol-3-monophosphate (PI3P) and, through its FERM domain, binds the N-x-x-[Y/F]-motifs present in the cytosolic domains of cargo. Cargo binding displaces the intramolecular association between the autoinhibitory motif (star symbol) and the cargo-binding pocket of SNX17 FERM domain. The non-autoinhibited cargo-bound SNX17 is then able to associate with the 16-subunit Commander assembly through the interaction between the carboxy-terminal IGDEDL[470] motif of SNX17 and the Retriever sub-assembly. In a process yet to be fully elucidated, Commander engages the WASH complex to control the dynamic turnover of endosomal actin. Actin branching supports the formation of cargo-loaded tubulo-vesicular carriers that transport cargo to their destination. Here, we propose that SNX17 may additionally protect cargoes, destined for the melanosome compartment, from lysosomal degradation. These cargoes may include regulators of melanosome neutralisation, a process that is necessary for the proper functioning of melanogenic enzymes. For simplicity, other sorting complexes such as ESCPE-I and Retromer, and their cargo, are not shown. **B** In addition to canonical binding to SNX17, Commander binds to LRMDA in a SNX17-exclusive manner. We propose that in this process, in contrast with PI3P recognition by SNX17, active RAB32[50,77] integrates into organelle membranes through C-terminal geranylgeranylation. Membrane-anchored RAB32 then dynamically recruits LRMDA to melanosome membranes[17]. This interaction is stabilised upon LRMDA binding to Commander through a mechanism like SNX17: LRMDA employs a carboxy terminal IRDDQL[226] motif to act as an adaptor between RAB32 and the Retriever sub-assembly of Commander. This direct interaction is necessary to support LRMDA-RAB32-mediated function in melanosome biogenesis, however, future work is required to establish those processes controlled by this assembly and to understand the relationship of this pathway to known melanosome trafficking and biogenesis machinery such as BLOC1-3 and AP-1/3, and to the canonical SNX17-Commander pathway.

biogenesis. Our analysis is consistent with a recent study by Singla et al.[2], which identified LRMDA as an interactor of Retriever as defined through the in vitro binding of the LRMDA carboxy-terminus to Retriever. Through a parallel analysis of SNX17, we have established the plastic nature of the Commander assembly within SNX17-dependent and LRMDA-dependent steps of a complex organelle biogenesis pathway. Our discovery of this alternative Commander assembly extends the functional significance of this multiprotein complex for human health and disease beyond the confines of SNX17-mediated cell surface recycling of numerous cargo proteins.

A role for Commander in melanosome biogenesis is consistent with an unbiased genome-wide screen that identified twelve subunits of Commander, as well as LRMDA and RAB32, as being required for human pigmentation, and confirmed differential expression of several Commander complex components correlating with the level of skin pigmentation in samples from human donors[46]. The same study has also validated a major reduction in pigmentation in melanocytes upon the suppression of CCDC22 and COMMD3. Additionally, the importance of LRMDA for pigmentation in vivo has been demonstrated through the knock-down of LRMDA in zebrafish leading to fewer pigmented melanosomes and a decrease in overall organism pigmentation[18]. A previous study that identified the association between LRMDA and RAB32 also established that RAB32 is able to recruit LRMDA to RAB32-decorated membranes, and active RAB32 is known to associate with the cytosolic leaflet of several membrane-demarcated organelles[17,47]. Therefore, we propose that LRMDA functions as an adaptor to recruit Commander to organelles harbouring the active RAB32 compartment identity cue. Additionally, we propose that an alternative RAB38-LRMDA-Commander assembly exists, but our data suggest that LRMDA-RAB38 interaction might be of lower affinity, which would be consistent with yeast two-hybrid screening performed by Beyers et al.[17]. By establishing the molecular mechanism and functional importance for the direct interaction between RAB32/RAB38-LRMDA and Retriever, we have paved the way to a greater understanding of those precise events in melanosome maturation that are controlled by this assembly.

In our study, we have demonstrated that in agreement with the recruitment of the full Commander assembly by the LRMDA-RAB32 pathway, knock-out of either VPS35L or CCDC22 leads to an apparent decrease in melanocyte pigmentation, and increased endo-lysosomal retention of the melanosome marker TYRP1. Both phenotypes reveal a high degree of commonality with LRMDA and RAB32 knock-out cells. Despite this, the knock-out of Commander components leads to a much greater defect in melanosome biogenesis, with a near-complete loss of melanosome stages II, III and IV observed by electron microscopy. The disparity between phenotypes observed in RAB32 and LRMDA-deficient and Commander-deficient cells suggests that the Commander complex regulates other, LRMDA-independent, processes in melanosome biogenesis. Additional research is necessary to establish the precise participation of the Commander in the regulation of human pigmentation and to delineate between the direct involvement, and roles that may arise as a downstream consequence of perturbed cargo trafficking in the absence of Commander function in endosomal sorting. Here, we propose two possible and non-mutually exclusive mechanisms through which Commander may be involved in melanosome maturation in an LRMDA-independent fashion: (i) through the association with cargo adaptor SNX17 to regulate the compartment deacidification, and (ii) through WASH complex-mediated control of endosomal actin turnover.

To our knowledge, SNX17 has not previously been observed to regulate pigment production, however, our data demonstrates that SNX17-deficient melanocytes exhibit decreased melanin synthesis and suggest that SNX17 plays an important role in regulating melanosome pH. SNX17 is principally considered to function as a cargo adaptor that protects cargoes from lysosomal degradation and mediates their

trafficking to target compartments[1,2,4,15,48,49]. Therefore, we speculate that SNX17 could facilitate transport of cargoes, important for melanosome neutralisation, to the melanosome. This would imply that SNX17 mediates cargo sorting to a non-plasma membrane compartment, possibly in a cell-specific manner. Still, it is important to acknowledge that SNX17 regulates the cell surface presentation of over 100 diverse cargoes[4], and therefore, it may be that it is the loss of cell surface presentation of conventional SNX17 cargo that leads to observed phenotypes in SNX17 knock-out cells, or similarly, that in absence of SNX17 function, its cargo may be mis-delivered to melanosome and cause the melanosome acidification.

The genome-wide screen establishing proteins involved in pigmentation also identified the actin polymerising WASH complex[46]. This study also carried out immunoprecipitation of melanosome organelles and showed the presence of COMMD3, suggesting that in melanocytic cells a proportion of Commander localises to melanosomes. This validates previous work establishing a function for WASH in driving F-actin polymerisation during the constriction and fission of melanosomal tubules that bud from the melanosome to remove and recycle cargo, including VAMP7, as part of the melanosome maturation pathway[45]. A complex known to function in pigmentation called BLOC-3 has been described as a GEF that activates RAB32 and RAB38, and is believed to recruit these small GTPases to the melanosome membrane[50]. The tubules that emanate from melanosomes containing VAMP7 are also decorated with RAB38 and depend on BLOC-3 for their formation[51]. An interesting model could be envisioned in which RAB32/RAB38 recruit Commander via LRMDA association to regulate WASH and actin polymerisation during budding of tubules from melanosomes (Fig. 8B).

Importantly, the loss of Commander may broadly affect endosomal recycling and, consequently, melanosome biogenesis. Our previous research has identified that the Commander complex functions in the organisation of the retrieval subdomain, as the VPS35L knock-out leads to loss of segregation between endosomal subdomains – a process that is independent of SNX17-mediated cargo sorting[1]. Within the endosomal network, Commander associates with the WASH complex where, through a mechanism yet to be elucidated, it controls and restricts the WASH-mediated formation of F-actin within the retrieval subdomain[34]. Singla et al.[34] also demonstrated that suppression of Commander results in a shift in the dynamic control of actin assembly and turnover leading to an overall accumulation of F-actin – a phenotype also observed in our study – and speculated that at the functional level, this perturbation in F-actin dynamics leads to inefficient cargo recycling. Indeed, consistent with the joint role in the control of actin dynamics, recent analysis has also shown that the WASH complex and Commander have co-evolved[52]. It is important to note that localised actin polymerisation, mediated by the WASH complex, has been implicated in regulating compartment acidification and neutralisation through the trafficking of the V-ATPase[53,54] which could contribute to altered melanosome acidification that we have observed in this study and that was reported in melanocytes lacking COMMD3 or CCDC22[46]. While we propose that one of the mechanisms through which Commander broadly regulates endosomal sorting to melanosomes is dependent on the control of WASH-mediated actin polymerisation on endosomes, this function might be complementary to Commander-mediated regulation of endosomal phosphoinositide turnover[34,55].

In this study, we use a melanocyte cell model to exemplify the functional importance of the RAB32-LRMDA-Commander assembly, however, it is important to note that LRMDA and RAB32 are expressed in many non-melanogenic cell types[17]. Indeed, RAB32 participates in several processes outside of the trafficking to melanosomes: it regulates the recycling of autophagosome components[56], controls mitochondrial dynamics[57], and broadly regulates the biogenesis of cell-type specific lysosome-related organelles (other than melanosomes) in

osteoclasts during bone homoeostasis[58,59], and dense granules in megakaryocytes[60] (reviewed in Bowman et al.[61]). RAB32 is also recruited to pathogen-containing vacuoles as part of a host defence mechanism during infection with *Salmonella*, *Legionella*, or *Trypanosoma*; this relies on RAB32 orchestrating the delivery of the mitochondrial metabolite itaconate to the vacuolar lumen[62–67]. Similarly, LRMDA has been implicated in a variety of disease conditions that appear to be un-related to melanosome biogenesis: a SNP (rs10824307) near LRMDA is associated with mild-obesity-related diabetes[68], and as highlighted previously[17], at least two OCA7 patients harbouring LRMDA mutations have recurrent infections[69]. SNPs within an intronic region of *LRMDA* are also associated with increased susceptibility to COVID-19 in non-Europeans and with platelet aggregation phenotypes[70], consistent with a broader role of LRMDA in immune cells and megakaryocytes. Several independent screens have also identified the involvement of Commander components in pathogen entry and phagocytosis[71–73], however, further extensive analysis is required to conclude whether LRMDA and Commander are involved in other diverse functions of RAB32.

Lastly, in this study, we used our mechanistic knowledge of SNX17 binding to Retriever to inform the search for interactors containing similar Retriever-binding sequences. This identified LRMDA as an adaptor, however, a less stringent search, updated with our knowledge of LRMDA-Retriever association, might reveal other possible Retriever-binding proteins. As we have shown here on the example of LRMDA-Commander, the identification of additional adaptors expands our understanding of Commander-mediated function outside of the canonical trafficking to the plasma membrane and offers the possibility that the Commander complex can mediate trafficking to other compartments.

## Methods

### Antibodies and reagents

Alexa Fluor 647 Phalloidin was purchased from Cell Signalling Technology (8940), Bafilomycin A1 from ApexBio (A8627), and LysoTracker Red DND99 from Invitrogen (L7528).

Primary antibodies against the following targets were used: CCDC22 (Proteintech, 16636, WB 1:1000), CCDC93 (LSBio, C336997, WB 1:1000), EEA1 (BD Biosciences, 610457, IF 1:400), FAM21 (Gift from Dan Billadeau, IF 1:1000), GFP (Roche, 11814460001, WB 1:2000), LAMP1 (Cell Signalling Technology, 9091, IF 1:400), LRMDA (Sigma, HPA050419, WB 1:500), LRP1 (Abcam, ab92544, WB 1:1000), mCherry (Antibodies.com, A85306, WB 1:1000), PMEL HMB45 (Dako/Agilent, M0643, WB 1:3000), RAB32 (GeneTex, GTX130477, WB 1:1000), SNX17 (Abcam, ab307513, WB 1:1000), TYRP1 (Abcam, ab3312, IF 1:400), VPS26C (Millipore/Sigma Aldrich, ABN87, WB 1:750), VPS29 (Santa Cruz, SC-398874, WB 1:200), VPS35L (Abcam, ab97889, WB 1:1000), β-actin (Sigma, A1978, WB 1:1000), RAB38 (Santa Cruz, sc-390176, WB 1:500), N-cadherin (Cell Signalling technology, 14215, WB 1:1000).

For secondary detection the following antibodies were used: Goat anti-Mouse IgG (H + L) Cross-Adsorbed Secondary Antibody, Alexa Fluor 680 (Invitrogen, A-21057, WB 1: 20000), Goat anti-Rabbit IgG (H + L) Secondary Antibody, DyLight 800 (Invitrogen, SA5-35571S, WB 1:20000), Goat anti-Rabbit IgG (H + L) Secondary Antibody, DyLight 800 (Cell Signalling Technology, 5151, WB 1:20000), Alexa Fluor 488 anti-mouse IgG (Invitrogen, A21202, IF 1:400), Alexa Fluor 568 anti-rabbit IgG (Invitrogen, A10042, IF 1:400).

### Cell lines and CRISPR KO

RPE1-hTERT, HeLa and HEK-293T cells were cultured in DMEM (Sigma Aldrich, D5796) with 10% FBS (Sigma Aldrich, F7524) and 1% penicillin/streptomycin (Gibco, 15140122) and MNT1 cells were grown in RPMI-1640 (Gibco, 11875093) medium with 20% FBS, 10% AIM-V (Gibco, 12055091), supplemented with 1% sodium pyruvate (Sigma Aldrich,

S8636) and 1% non-essential amino acids (Gibco, 11140050). All cell cultures were maintained at 37 °C with 5% $CO_2$.

RPE1-hTERT, HeLa and HEK-293T cells were sourced from ATCC, and MNT1 parental and LRMDA KO cell line was a gift from Prof. Santiago di Pietro. HeLa VPS26C CRISPR KO cell line was previously established in the laboratory.

To establish CRISPR KO MNT1 cells, MNT1 cells were seeded into 6-well plates the day prior to transfection and were transfected at 25% confluency. Lipofectamine LTX reagent (Thermo Fisher Scientific, 15338030) was used to introduce the CRISPR gRNA-expressing pX459 plasmid according to the manufacturer's instructions. Briefly, 1μg of DNA was mixed with 3.5 μl of Lipofectamine LTX Plus reagent in 100 μl of OptiMEM (Gibco, 31985-070), and in a separate tube, 3.5 μl of Lipofectamine LTX reagent was diluted in another 100 μl of OptiMEM. The content of the two tubes was then mixed, and incubated at room temperature for 5 min, and the DNA-OptiMEM mixture was then added to MNT1 cells in fresh growth medium. After 36 h, the transfected cells were selected with 3 μg/ml puromycin for 24 h. After selection, cells were left to recover for 3 days in normal growth media before being plated into 96 wells for clonal selection. After 2 weeks, wells containing a single colony were expanded into larger dishes and CRISPR KO was validated using western blot.

The CRISPR/Cas9 sequence targeting RAB32 was designed using CRISPick tool, the targeting sequence was 5′GGTGATCGGC-GAGCTTGGCG (exon 1). Other sgRNA sequences were designed using CRISPRdirect. The following target sequences were used: VPS35L – 5′ CCTGTTTCTTGTTCGAGAGCTTC (exon 28), SNX17 – 5′CCTTCGG AGTCAAGAGTATAAGA (exon 7), CCDC22 – 5′CCTCGCTCCTCGAACA CCATGCC (exon 6), FAM21 – 5′CCCCTATGGTCCCCCTCCCACTG (exon 15) and VPS26C – 5′CGTGGTGGTCATATCGAGTAAGG (exon 2).

### Molecular cloning and site-directed mutagenesis (SDM)

LRMDA-GFP, LRMDA-GFP(ΔC) and LRMDA-GFP(ΔC) in pEGFP-N3 plasmid were a gift from Prof. Santiago di Pietro, and the wild-type sequence encoded the longer LRMDA isoform with Uniprot accession A0A087WWI0. This isoform was used in all experiments. To transfer the sequences into pEGFP-C1, LRMDA sequence was amplified in PCR reaction, the amplified bands were then resolved on 1% agarose gel and purified using GFX PCR DNA and gel band purification kit (Cytiva, 28903471). The amplified fragments and empty pEGFP-C1 were digested with EcoRI (NEB, R3101) and SalI (NEB, R3138), and the plasmid backbone was additionally treated with quickCIP to prevent auto-ligation. The digestion reactions were purified using DNA purification kit and used for ligation reaction with T4 ligase (NEB, M0202). The ligation reaction was then transformed into competent XL-Gold ultracompetent bacteria (Agilent, 200315). After colony PCR using HotStart Q5 2x mastermix (NEB, M0494), colonies positive for insertion were used for DNA isolation using MiniPrep (QIAprep Spin Miniprep Kit, Qiagen, 27106) and MaxiPrep (HiSpeed Plasmid Maxi Kit, Qiagen, 12663) and finally, the correct sequences were validated by Sanger sequencing of the whole open reading frame. A similar protocol was followed to transfer wild-type or mutant GFP-LRMDA and GFP-VPS26C sequences into the pLVX plasmid for lentiviral expression, but the amplified bands and pLVX backbone were digested using XmaI (NEB, R0180) and XbaI (NEB, R0145).

For site-directed mutagenesis, the primers were designed using the Agilent primer design tool, and the QuickChange II kit (Agilent, 200523-S) was used to introduce desired mutations according to the manufacturer's instructions. Briefly, 50 ng of template DNA were used in SDM reactions with 125 ng of each SDM primer, 1 μl of dNTP mix and 1 μl of PfuUltra HF DNA polymerase (2.5 U/μl) in 1x reaction buffer (10-fold dilution) in ddH2O. The sequences were amplified to introduce desired modification using the following conditions: 1 cycle at 95 °C for 30 seconds, followed by 16 cycles of incubations at 95 °C for

30 seconds, 55 °C for 1 minute and at 68 °C for 1 minute per kb of template. Non-amplified DNA was removed through DpnI digestion at 37 °C for 1 h. 5 μl of the digestion reaction was transformed into either XL1-Blue competent (Agilent, 200249) or XL10-Gold ultracompetent cells and plated on LB agar with the appropriate antibiotic. Selected colonies were then used for MiniPrep and the specific and correct introduction of mutation was validated using Sanger sequencing of full open reading frame.

Sequences of primers used in this study are listed in Supplementary Data 2.

### Recombinant protein purification

GST-RAB32 and HaloTag-LRMDA encoding-genes, with TEV cleavage site between the N-terminal tag and human RAB32 or LRMDA, were synthesised by Twist Biosciences (San Francisco, CA), using codon optimisation for S. frugiperda. GST-RAB32 and Halo-LRMDA sequences were subcloned into pACEBac1 backbone after digestion with BamHI and HindIII. Baculoviruses were then prepared as described in Healy et al[6]., using pACEBac1-GST-TEV-RAB32 and pACEBac1-HaloTag-TEV-LRMDA. For protein expression, 4 ml of V1 baculovirus was used to infect 400 ml of Sf21 cells ($1 \times 10^6$ cells/ml) in Sf-900 II SFM media (Thermo, 10902104). Cells were observed under an inverted fluorescence wide-field microscope for protein expression, and the pellets were collected 3 days after transfection and stored at −80 °C. For protein purification, cells were lysed on ice by sonication (130-Watt Ultrasonic Processor, Cole-Palmer) using 10 s sonication intervals with 30 s 'off' cycles for 150 s, in lysis buffer (50 mM HEPES pH 7.2, 150 mM NaCl, 2 mM β-mercaptoethanol, 0.1% (v/v) Triton X-100) with added protease inhibitors. Next, lysates were clarified by centrifugation at 20000 RPM (JA-25 fixed-angle rotor, Beckman Coulter) at 4 °C. Supernatants containing GST-TEV-RAB32 or HaloTag-TEV-LRMDA were then collected and added to pre-washed Glutathione Sepharose 4b beads (Cytiva, 17075601) or HaloLink resin (Promega, G1914). The samples were incubated for 1 h at room temperature, and following the incubation, beads were collected by centrifugation at 350 RCF for 3 min at 4 °C, and beads were washed thrice in 10 ml lysis buffer. To generate tag-less RAB32 and LRMDA, beads were then incubated with TEV protease (NEB, P8112) in 3 ml elution buffer (50 mM HEPES pH 7.2, 150 mM NaCl, 2 mM β-mercaptoethanol) with 1 mM DTT and protease inhibitors. Samples were collected from all stages of purification to evaluate purification efficiency.

The eluates were collected and concentrated using Amicon® Ultra Centrifugal Filter with 10 kDa molecular weight cut-off (Sigma, UFC801024), and aggregated protein was removed by centrifugation of concentrated protein lysate at 16000 RCF for 5 min at 4 °C. The supernatants were aliquoted, flash-frozen in liquid nitrogen and kept at −80 °C until use.

Retriever with His-VPS29 was purified as described in Healy et al.[6].

### In vitro LRMDA-RAB32-Retriever interaction assay

For in-vitro pull-downs, TALON Superflow beads (Cytiva, 28957499) were washed thrice in binding buffer (50 mM HEPES pH 7.2, 150 mM NaCl, 2 mM β-mercaptoethanol and 0.1% (v/v) Triton x-100) with added protease inhibitors. 100 μl of binding buffer with no added protein, or combinations of 1 μM LRMDA, 1 μM RAB32 and 0.5 μM Retriever with VPS29-His tag were then added to the beads. Samples without added Retriever were used to evaluate nonspecific binding of RAB32 and LRMDA to beads (background). The samples with beads were then resuspended on ice, and 30 μl of the sample was collected for input. The remaining samples were incubated rotating at 4 °C for 45 min. After the incubation, beads were collected by centrifugation at 350 RCF for 90 s at 4 °C. The beads were washed thrice with 300 μl of His-tag wash buffer (50 mM HEPES pH 7.2, 150 mM NaCl, 2 mM β-mercaptoethanol, 10 mM imidazole and 0.1% (v/v) Triton-x100) with added protease inhibitors. Input samples were mixed with equal volume of 4x NuPage LDS sample buffer with 2.5% β-mercaptoethanol (Invitrogen, NP0008), and beads with equal volume of 2x NuPage LDS sample buffer with 2.5% β-mercaptoethanol.

### Transfection and GFP-trap experiments

For GFP trap experiments using HeLa and HEK293T cells, cells were grown in 150 mm dishes and transfected the day after plating at 90–95% confluency. To prepare the transfection reaction, 15 μg of plasmid DNA were diluted in 5 ml of OptiMEM and in a separate tube of 5 ml OptiMEM, PEI was diluted at 3:1 PEI-DNA ratio. The two dilutions were mixed at 1:1 ratio and left to incubate at room temperature for 15 min. Then, the growth media was aspirated from cell dishes and replaced for the OptiMEM PEI-DNA mixture. Cells were incubated with the mixture for 6 h, after which the transfection mixture was replaced for normal growth media. The cells were then left to grow for 24 h before harvesting.

For GFP-trap experiments, cells were inspected for GFP signal under inverted fluorescence microscope and then cell culture media was removed, and the cells were washed twice with ice-cold PBS. Cells were then lysed in lysis buffer (0.5% NP-40, 100 mM Tris-HCl pH 7.5 in PBS, or in 0.5% NP-40, 100 mM Tris-HCl pH 7.5 in dH2O for experiments involving GFP-SNX17 or stable cell lines) and lysates were clarified by centrifugation at 20000 RCF at 4 °C for 10 min. Input samples were collected after centrifugation and resuspended with equal volume of 4x NuPage LDS sample buffer with 2.5% β-mercaptoethanol. Next, supernatants were added to 25 μl of pre-washed GFP-Trap agarose beads (Chromotek, gta-20). The samples were incubated rotating at 4 °C for 1 h. After incubation, the pulldown reactions were washed by pelleting the beads at 350 RCF at 4 °C for 1 minute, and resuspending in first wash buffer (0.25% NP-40, 100 mM Tris-HCl pH 7.5 in PBS, or in 0.5% NP-40, 100 mM Tris-HCl pH 7.5 in dH2O for experiments involving GFP-SNX17 or stable cell lines), repeating the pelleting and wash step twice. Next, they were washed in second wash buffer (100 mM Tris-HCl pH 7.5 in PBS, or in 0.5% NP-40, 100 mM Tris-HCl pH 7.5 in dH2O for experiments involving GFP-SNX17 or stable cell lines) and the beads were again collected by pelleting at 350 RCF at 4 °C for 1 minute. Lastly, the wash buffer was aspirated and the beads were resuspended in 30 μl of 2x NuPage LDS sample buffer with 2.5% β-mercaptoethanol. All samples were denatured at 95 °C for 10 min.

### Lentiviral particle production and generation of stable cell lines

For lentiviral particle production, HEK-R293T cells were plated the day prior to transfection, and transfected at 80% confluency. 20 μg of pLVX, 15 μg of pAX2, 5 μg of pMD2.G were diluted in 7 ml of OptiMEM. A separate tube was used to dilute PEI in 7 ml of OptiMEM at 1:3 DNA:PEI ratio. The two mixtures were mixed at 1:1 ratio and left to incubate for 15 min at room temperature. Cells were incubated with the mixture for 4 h, after which the transfection mixture was replaced for normal growth media. The cells were then left to grow for 48 h, after which the media was collected, centrifuged at 3000 RCF for 10 min and filtered through 0.45 μm filter.

For generation of stable cell lines, cells were plated into 6-well dishes and transduced on the same day at 20% confluency with dropwise addition of lentiviral media. After 48 h, transduced cells were selected with puromycin (3 μg/ml for MNT1 and 20 μg/ml for RPE1-hTERT) until all untransduced cells in control well were detached.

### TMT proteomics for interactome analysis

Stably transduced cell lines were used for GFP-trap experiments as described above, and samples on beads were submitted for proteomic analysis. For the analysis of interactomes in RPE1-hTERT cell line, each experiment was performed in three biological replicates, with two technical repeats. Two technical replicates from experiments shown in Fig. 1E were removed from the statistical analysis, as the principal component analysis identified two outliers, defined by their

comparatively lower abundance of protein in wild-type GFP-LRMDA samples. The remaining dataset still contained three biological replicates. The validation of interactomes in MNT1 cell line was carried out in three biological replicates.

**TMT Labelling and High pH reversed-phase chromatography.** Samples were reduced (10 mM TCEP, 55 °C for 1 h), alkylated (18.75 mM iodoacetamide, room temperature for 30 min) and digested from the beads with trypsin (1.25 µg trypsin; 37 °C, overnight). The resulting peptides were labelled with TMT 10plex or TMTpro16Plex reagents according to the manufacturer's protocol (Thermo Fisher Scientific, Loughborough, LE11 5RG, UK) and the labelled samples pooled and desalted using a SepPak cartridge according to the manufacturer's instructions (Waters, Milford, Massachusetts, USA). Eluate from the SepPak cartridge was evaporated to dryness and resuspended in buffer A (20 mM ammonium hydroxide, pH 10) prior to fractionation by high pH reversed-phase chromatography using an Ultimate 3000 liquid chromatography system (Thermo Scientific). In brief, the sample was loaded onto an XBridge BEH C18 Column (130 Å, 3.5 µm, 2.1 mm × 150 mm, Waters, UK) in buffer A and peptides eluted with an increasing gradient of buffer B (20 mM Ammonium Hydroxide in acetonitrile, pH 10) from 0–95% over 60 min. The resulting fractions were evaporated to dryness and resuspended in 1% formic acid prior to analysis by nano-LC MSMS using an Orbitrap Fusion Tribrid mass spectrometer (Thermo Scientific).

**Nano-LC mass spectrometry.** High pH RP fractions were further fractionated using an Ultimate 3000 nano-LC system in line with an Orbitrap Fusion Tribrid mass spectrometer (Thermo Scientific). In brief, peptides in 1% (vol/vol) formic acid were injected onto an Acclaim PepMap C18 nano-trap column (Thermo Scientific). After washing with 0.5% (vol/vol) acetonitrile 0.1% (vol/vol) formic acid, peptides were resolved on a 250 mm × 75 µm Acclaim PepMap C18 reverse phase analytical column (Thermo Scientific) over a 150 min organic gradient with a flow rate of 300 nl min−1. Solvent A was 0.1% formic acid and Solvent B was aqueous 80% acetonitrile in 0.1% formic acid. Peptides were ionised by nano-electrospray ionisation at 2.0 kV using a stainless-steel emitter with an internal diameter of 30 µm (Thermo Scientific) and a capillary temperature of 275 °C.

All spectra were acquired using an Orbitrap Fusion Tribrid mass spectrometer controlled by Xcalibur 2.1 software (Thermo Scientific) and operated in data-dependent acquisition mode using an SPS-MS3 workflow. FTMS1 spectra were collected at a resolution of 120,000, with an automatic gain control (AGC) target of 200,000 and a max injection time of 50 ms. Precursors were filtered with an intensity threshold of 5000, according to charge state (to include charge states 2-7) and with monoisotopic peak determination set to peptide. Previously interrogated precursors were excluded using a dynamic window (60 s +/−10 ppm). The MS2 precursors were isolated with a quadrupole isolation window of 1.2 m/z. ITMS2 spectra were collected with an AGC target of 10,000, max injection time of 70 ms and CID collision energy of 35%.

For FTMS3 analysis, the Orbitrap was operated at 50,000 resolution with an AGC target of 50,000 and a max injection time of 105 ms. Precursors were fragmented by high energy collision dissociation (HCD) at a normalised collision energy of 60% to ensure maximal TMT reporter ion yield. Synchronous Precursor Selection (SPS) was enabled to include up to 10 MS2 fragment ions in the FTMS3 scan. The tolerance for reporter ion detection was set at 20 ppm in the Reporter Ions Quantifier node of the Proteome Discoverer software.

**Data analysis.** The raw data files were processed and quantified using Proteome Discoverer software v2.4 (PD2.4, Thermo Scientific) and searched against the UniProt Human database (downloaded January 2024: 82415 entries), an in-house contaminants database, and the GFP sequence using the SEQUEST HT algorithm. The master protein selection was improved with an in-house script which searches Uniprot for the current status of all protein accessions and updates redirected or obsolete accessions. The script further takes the candidate master proteins for each group, and uses current uniprot review and annotation status to select the best annotated protein as master protein without loss of identification or quantification quality. Peptide precursor mass tolerance was set at 10 ppm, and MS/MS tolerance was set at 0.6 Da. Search criteria included oxidation of methionine (+15.995 Da), acetylation of the protein N-terminus (+42.011 Da) and methionine loss plus acetylation of the protein N-terminus (−89.03 Da) as variable modifications and carbamidomethylation of cysteine (+57.021 Da) and the addition of the TMT mass tag (+229.163 Da for experiments using 10Plex tags or +304.207 Da for experiments using 16Plex tags) to peptide N-termini and lysine as fixed modifications. Searches were performed with full tryptic digestion and a maximum of 2 missed cleavages were allowed. The following thresholds were applied in the Reporter Ions Quantifier Node of the Proteome Discoverer software: Co-isolation threshold:75; Average Reporter S/N threshold: 10, SPS Mass Matches Threshold: 65. The reverse database search option was enabled and all data was filtered to satisfy false discovery rate (FDR) of 5%.

All data processing and statistical analysis outside of PD2.4 was performed in the R statistical environment (v4.2). Where appropriate and indicated, protein abundances for each sample in an experiment were normalised such that all samples had an equal total PSM abundance. Abundances were Log2 transformed to bring them closer to a normal distribution prior to statistical analysis. Differential protein abundance analysis was performed using univariate paired t-tests. For all comparisons, the p-value was adjusted using the Benjamini-Hochberg FDR method. PCAs were calculated using the FactoMineR package, and the plotted using the ggplot2 package.

## Cell surface protein biotinylation

The cell surface biotinylation was performed as described previously in Butkovič et al.[1]. MNT1 cells were placed on ice and washed with ice-cold PBS prior to labelling with cell-impermeable 0.2 mg/mL Sulfo-NHS-SS Biotin (ThermoFisher Scientific, no. 21217) in PBS (pH 7.4). After 30 min of biotinylation, the cells were washed with TBS and the biotinylation reaction was quenched by incubating the cells in TBS for 10 min, The cells were kept on ice during the biotinylation, quenching and washing steps to prevent endocytosis and unspecific labelling of intra-cellular proteins.

After quenching, cells were washed with PBS and lysed in lysis buffer containing 2% triton x-100 with protease inhibitors in PBS. The lysates were clarified by centrifugation at 16000 RCF for 10 min. Supernatants were collected and total protein levels were analysed using BCA reaction. The samples were next diluted to equal protein concentrations, and inputs were collected. Equal amounts of protein were incubated with Streptavidin beads (GE Healthcare, USA) for 30 min at 4 °C. The beads were pelleted by centrifugation at 350 RCF for 1 minute. The beads were first washed in PBS with 1% triton x-100, twice in PBS with 1% triton x-100 with 1 M NaCl and once with PBS. Between the washes, the beads were collected at the bottom of the tube by 1 min centrifugation at 350 RCF. Finally, the beads were resuspended in a 2x loading buffer, containing 2.5% β-mercaptoethanol and all samples, including inputs, were denatured at 95 °C for 10 min.

## Western blot

For analysis of whole-cell lysates, cells were grown in 6-well dishes, growth media was aspirated and the cells were washed with ice-cold PBS. Then, the cells were lysed with triton lysis buffer (1.5% x-100 Triton in PBS with protease inhibitors) or RIPA buffer (150 mM NaCl, 1% NP40, 0.5% sodium deoxycholate, 0.1% SDS, 50 mM TRis) with

protease inhibitors for PMEL fragment analysis by HMB45 antibody. Lysates were clarified by pelleting at 16000 RCF at 4 °C for 10 min. Supernatants were transferred into fresh tubes and protein concentration was measured for each sample using Pierce BCA protein assay kit (Thermo Scientific, 23227). The samples were then diluted to equal concentrations in lysis buffer and 4x NuPage LDS sample buffer with 2.5% β-mercaptoethanol was added at 1x final concentration. Samples were then denatured at 95 °C for 10 min.

Protein lysates were resolved on 4–12% Bis-Tris NuPAGE (Thermo Fischer, NP0336) precast gels. The proteins were next transferred onto PVDF membranes (Cytiva, 10600023) for 75 min at 100 V in ice-cold transfer buffer (10% methanol, 25 mM tris and 192 mM glycine in mqH$_2$O). PVDF membranes were covered with 10% milk in TBST (tris-buffered saline with Tween, containing 150 mM NaCl, 10 mM tris pH7.5, 0.1% Tween-20) for 1 h at room temperature for blocking. The membranes were rinsed multiple times in TBST and incubated overnight in primary antibody diluted in 3% BSA in TBST. Following the primary antibody incubation, the membranes were washed three times in TBST and incubated for 1 h with 1:20000 dilution of appropriate secondary antibodies for 1 h at room temperature. The membranes were then washed three times with TBST and rinsed with mqH$_2$O before imaging on Odyssey Clx scanner (LICORbio).

## Melanin content measurement
Cells were cultured in 100 mm dishes for three days and harvested at 90% confluency to perform melanin content measurement. The media was aspirated, and cells were washed with DPBS, before trypsinisation. Trypsin was then inactivated with growth media containing FBS. The cells were pelleted at 400 g for 5 min and washed twice with cold DPBS, before resuspension in lysis buffer (1.5% (v/v) Triton X-100 in PBS with protease inhibitors). Lysates were then sonicated for 4 min in water bath, and then pelleted at 20000 RCF for 15 min at 4 °C. Supernatant was removed and used for BCA protein content assay. The pellets were washed with 100% ethanol, vortexed, and then pelleted again at 20000 RCF for 5 min at 4 °C. All ethanol was aspirated, and pellets (tubes open) were left to dry at 37 °C for 30 min. 600μl of freshly prepared 10% DMSO (v/v) in 1M NaOH in PBS was then added to the pellets, and the samples were incubated in a heating block at 80°C for a total duration of 1h, vortexing the samples after 10min of incubation and again at the end of the incubation. Samples were then left at room temperature for 5 min to cool down, and centrifuged at 20000 RCF for 10 min. 200 μl of each sample was transferred into a 96-well plate and absorbances were measured on POLARstar Omega microplate reader (BMG LABTECH) at 490 nm. The average absorbance per sample was then normalised to protein concentration, analysed by BCA assay. Lastly, all melanin content measurements were compared to the parental MNT1 cell line melanin content.

For experiments involving bafilomycin (and corresponding DMSO control) the same protocol was used with the following modifications: cells were grown in 65 mm plates, and after drying at 37 °C, the pellets were resuspended in 300 μl of freshly prepared 10% DMSO (v/v) in 1 M NaOH in PBS.

Representative photos of pellets were acquired before lysis.

## Fluorescence microscopy sample preparation and image acquisition
MNT1 cells were grown on coverslips for 2 days prior to fixation to allow proper attachment. The cells were fixed for 15 min with 4% PFA in PBS. Fixative was aspirated and the coverslips were rinsed thrice with DPBS, before permeabilisation with 0.1% Triton x-100 in PBS or 0.1% saponin in PBS (for TYRP1 and LAMP1 co-staining). Coverslips were washed with PBS three times, and then blocked with 1% BSA in PBS for 30 min. Coverslips were then rinsed with DPBS and incubated with appropriate primary antibodies, diluted in 0.1% BSA in PBS or 0.1% BSA with 0.01% saponin in PBS (for TYRP1 and LAMP1 co-staining) for 1 h at

room temperature. After primary antibody staining, the coverslips were again washed thrice with PBS and stained with appropriate secondary antibodies, diluted in 0.1% BSA in PBS or 0.1% BSA with 0.01% saponin in PBS (for TYRP1 and LAMP1 co-staining) for 1 h at room temperature. The coverslips were then washed twice in PBS and rinsed once in mqH$_2$O before mounting on slides using Fluoromount G mounting medium (Invitrogen, 00495802).

Cells were grown on coverslips for 2 days prior to LysoTracker labelling. Cell culture medium was aspirated and replaced with fresh growth medium containing 100nM Lysotracker DND-99, and cells were incubated at 37°C, 5% CO2 incubator for 30min. The media was then aspirated, and cells were briefly washed with generous amount of DPBS, before 10-minute fixation with 4% PFA in PBS. Coverslips were then washed with DPBS and incubated for 5 min with 1:10000 dilution of DAPI in PBS, washed twice with PBS and rinsed once with mqH$_2$O before mounting on microscopy slides using Fluoromount G mounting medium. The Lysotracker-labelled cells were analysed on the confocal microscope within 24 h of staining.

Samples were analysed on Leica SP8 AOBS confocal laser scanning microscope attached to a Leica DM I8 inverted epifluorescence microscope (Leica microsystems), using 63×1.4 NA oil immersion objective (Leica, 506350) using Leica LAS software. For acquisition 2x zoom was used and z-stacks were acquired.

## Live imaging
For live imaging experiments, MNT1 cells were seeded into 35 mm glass bottom dishes (Mattek) and transfected using FuGene 6 Transfection reagent 24 h prior to image acquisition.

Images were acquired using an Olympus IX83 inverted microscope, equipped with a Yokogawa CSU-W1 SoRa spinning disk and Hamamatsu Fusion BT sCMOS cameras. During the image acquisition, the cells were kept at 37 °C in an environmental chamber (PECON) with temperature control and CO$_2$ enrichment.

Olympus CellSens imaging software was used to acquire single z-slices, using 60× 1.5 NA oil immersion objective.

## Transmission electron microscopy
MNT1 parental, LRMDA KO, and VPS35L KO cells were fixed in a mixture of 2.5% glutaraldehyde and 2% paraformaldehyde in 0.1M PHEM buffer, pH 6.9, at room temperature for 2h. Afterwards, cells were rinsed and stored in 1% formaldehyde in 0.1 M PHEM Buffer at 4 °C. The cells were poststained with 1% osmium tetroxide, 1.5% potassium ferrocyanide in PHEM for 2 h at 4 °C. Samples were then dehydrated using a graded aceton series and infiltrated with EMbed resin (Electron Microscopy Sciences). Resin was polymerised for >48 h at 65 °C. After polymerisation, sample blocks were trimmed, and 70 nm sections were cut and collected on formvar and carbon-coated copper TEM grids (Cell Microscopy Core, UMC Utrecht). Sections on grids were stained with uranyl acetate and lead citrate (Leica AC20). Micrographs were collected on a JEM1011 (JEOL) equipped with a Veleta 2k × 2k CCD camera (EMSIS) or on a Tecnai12 (Thermo Fisher) equipped with a Veleta 2k × 2k CCD camera (EMSIS) and operating SerialEM software[74].

## AlphaFold2 modelling
All AlphaFold2 models presented in this study were generated with the ColabFold implementation of AlphaFold multimer version 3[75]. Amber relaxation was applied for modelling of RAB32-LRMDA-Retriever to resolve the position of side chains. For each prediction, 5 models (ranks) were generated, and we assessed their quality through ipTM scores, pLDDT scores and predicted alignment error plots.

## Data analysis and data presentation
ipTM scores from AlphaFold2 modelling of [IL]-x-[DE]-x-x-[IL]> sequences against VPS35L:VPS26C were imported into R studio. The

'per-chain' ipTM score for each ScanProsite hit was then plotted for all 5 ranks using ggplot package.

All AlphaFold2 models were opened in UCSF ChimeraX software, which was used to visualise structural models.

Data obtained after statistical analysis of proteomic analysis were opened in R studio and volcano plots were prepared using EnhancedVolcano package.

Confocal microscopy images were analysed in Volocity 6.3.1 software (PerkinElmer), which was used to determine Pearson's colocalisation coefficients with automatic Costes background thresholding. Representative images for colocalisation images were also prepared using the Volocity software.

Electron microscopy images were analysed using Fiji software. For quantification, stitched microscopy images were used. Images were randomised prior to analysis of melanosome stages.

For western blot analysis, band intensities were analysed in Image Studio Lite software (LICORbio).

GraphPad Prism 8 was used for statistical analysis and graph preparation, and information regarding statistical tests and n-number is reported in figure descriptions.

### Statistics and reproducibility

Unless otherwise stated in the figure legends, all experiments were performed in at least three independent biological replicates, and representative data are shown. Blinding was used for analysis of electron microscopy images.

### Reporting summary

Further information on research design is available in the Nature Portfolio Reporting Summary linked to this article.

## Data availability

All data and reagents will be made available upon request. Source Data are provided with this paper. The mass spectrometry proteomics data have been deposited to the ProteomeXchange Consortium via the PRIDE[76] partner repository with the dataset identifier PXD067193. The AlphaFold models presented in this study were deposited onto ModelArchive database and are available under accession numbers ma-j7h7g (Retriever-LRMDA-RAB32), ma-2bezz (LRMDA-RAB32) and ma-f3hs4 (LRMDA-RAB38). Source data are provided with this paper.

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

## Acknowledgements

We thank the Wolfson Bioimaging Facility at the University of Bristol for their support. R.B. is supported by the EndoConnect European Research Training Network (No. 953489). M.D.H. is supported by a Dementia Australia Research Foundation (DARF) project grant. K.E.M. is supported by the Wellcome Trust through a Sir Henry Wellcome Postdoctoral

Fellowship (220480/Z/20/Z). N.L. is supported by a Dutch Research Council (NWO) grant P21-30/4. N.L. and J.K. acknowledge the NWO National Roadmap for Large-Scale Research Infrastructure grants NL-Bioimaging (184.036.012) and NEMI (184.034.014). S.D.P. is supported by National Institutes of Health grant RO1HL151988 and National Science Foundation grant MCB-2313900. B.M.C. is supported by an Investigator Grant, Senior Research Fellowship, and Project Grant from the National Health and MRC (APP2016410, APP1136021, and APP1156493). Work in the Cullen laboratory is supported by the Wellcome Trust (104568/Z/14/Z and 220260/Z/20/Z), the Medical Research Council (MR/L007363/1 and MR/P018807/1), the Lister Institute of Preventive Medicine, and the award of a Royal Society Noreen Murray Research Professorship to P.J.C. (RSRP/R1/211004).

## Author contributions

R.B., M.D.H., B.M.C. and P.J.C. designed experimental concept; R.B. performed biochemical, insect cell recombinant protein expression, molecular and cellular analysis; A.P.W. assisted with insect cell expression; R.B. and M.D.H. AlphaFold2 modelling; W.B. and S.D.P. provided reagents and technical insight; K.E.M. obtained key preliminary data; C.deH., R.B., N.L. and J.K. performed EM experiments and ultrastructural analysis; P.A.L. and K.J.H. performed proteomics and proteomic analysis; S.D.P., B.M.C. and P.J.C. performed supervision and obtained funding; R.B. and P.J.C. wrote 1st draft with all authors contributing to the final manuscript.

## Competing interests

The authors declare no competing interests.
