## [Transparent Peer Review file · Nature Communications]

Identification of a RAB32-LRMDA-Commander membrane trafficking complex reveals the molecular mechanism of human oculocutaneous albinism type 7

Corresponding Author: Professor Peter Cullen

Version 0:

Reviewer comments:

Reviewer #1

(Remarks to the Author)

This manuscript identified LRMDA as a novel adaptor protein that links the Commander complex to active RAB32, elucidating a previously unrecognized pathway essential for melanosome biogenesis and pigmentation. The study provides mechanistic insights into the molecular basis of oculocutaneous albinism type 7 (OCA7), demonstrating how disease-associated mutations in LRMDA disrupt its interaction with RAB32 and the Commander complex. By revealing the mutually exclusive nature of LRMDA-Commander and SNX17-Commander assemblies, the authors highlight the functional plasticity of Commander beyond its canonical roles in endosomal recycling. The study establishes that LRMDA functions independently of SNX17, indicating distinct roles for these assemblies in organelle biogenesis pathways. Overall, the manuscript is well written and presents solid proteomic evidence for LRMDA's role in Commander assembly and melanosome biogenesis. Minor points are listed below, which should be addressed before this manuscript can be accepted for publication.

1. RAB32 and RAB38 are regulators of intracellular trafficking, influencing various physiological processes, including pigmentation. Although the authors did not observe interactions between RAB38 and LRMDA (Figure 1E), the close structural similarity of RAB38 to RAB32 suggests potential redundancy or compensatory mechanisms. Can the authors indicate if differences in protein expression account for these observations and if these mechanisms may be active in cell types beyond RPE1?
2. In the section 'Cooperativity in binding within the RAB32:LRMDA:Retriever complex', the last statement regarding the lack of mechanistic explanation may unintentionally downplay the significance of the proteomic findings. Expanding on this in the discussion would provide valuable insights. The observed cooperativity in binding between these proteins likely arises from at least three synergistic mechanisms: 1) LRMDA functions as an adaptor protein, linking the Retriever complex to RAB32. The carboxy-terminal domain of LRMDA directly associates with the Retriever complex via a conserved motif (IRDDQL226), specifically binding to VPS35L; 2) In its active GTP-bound state RAB32 favors its interaction with LRMDA, and facilitates the recruitment of the Commander complex to the RAB32-decorated membrane; 3) Mutual enhancement of binding: The association between LRMDA and Retriever enhances the binding of LRMDA to RAB32, while conversely, RAB32 binding strengthens LRMDA's association with the Retriever complex. This suggests a positive feedback loop, where the binding of one partner promotes the interaction of the others. This cooperativity likely ensures the formation of a stable and functional RAB32:LRMDA:Retriever complex.
3. For the proteomic data, the authors should deposit their raw data files in a publicly accessible repository, like PRIDE, to ensure transparency and reproducibility of their findings, allowing other scientists to access and analyze the raw data used in the study.
4. In the method section, the authors should specify the experimental workflow used in the tandem mass tag (TMT)-based quantitative proteomics, including the TMT labeling scheme, criteria for peptide and protein identification. They should clarify if all data reported for quantitation used SPS-MS3 and indicate mass tolerance applied to fragment ions (e.g. fragment ion tolerance was set to 0.9 Da for SPS-MS3). More information is required to assess how authors filtered PSMs with poor quality (e.g. spectra with TMT reporter summed signal-to-noise measurements that were less than 200, or with no MS3

spectra were excluded from quantitation).

5. The specific statistical methods employed for data analysis is not consistent between method and figure caption. For example, in the method they indicated p-value adjusted using the Benjamini-Hochberg FDR method, but Figure 1E only indicates significantly enriched proteins with $p < 0.01$ and Log₂ fold change > 2 are labelled, which appear arbitrarily defined. Clarification is also needed on the number of biological and technical replicates included to assess reproducibility and statistical power. For example, Figure 1E stated $n=6$, but there is no indication if these are biological or technical replicates.

6. Proteomics analyses identified LRMDA to accession #: A0A087WWI0 (226 AA) though another similar variant exists: Q9H2I8 (198 AA). Can the authors clarify if their mass spectrometry data allowed differentiation between these variants?

Reviewer #2

(Remarks to the Author)

Comments to the author:

Melanosome biogenesis follows sequential maturation of endocytic organelles into mature melanosomes in four stages. This process requires sequential transport of melanocytic cargo to form these unique organelles. Sorting endosomes (SE) and recycling endosomes (RE) play key roles during these processes. However, the role of early endosomal (EE) cargo sorting pathways in contributing the melanosome maturation is unknown. In this manuscript, the authors tested the role of one such sorting mechanism involving the Commander complex. The authors initiated their studies by identifying the LRMDA (leucine-rich melanocyte differentiation-associated) protein as an interactor to the Commander complex. Their *in vitro* modelling studies illustrated that LRMDA binds weakly (compared to SNX17) to the Commander complex in association with active Rab32. They used biochemical assays to prove the interaction between Rab32-LRMDA-Commander. Moreover, they have carried out gene knockouts (KO) in melanocyte cell line MNT1 and showed a decrease in melanocyte pigmentation. Moreover, the authors used pathogenic mutants of LRMDA that are associated with oculocutaneous albinism type 7 (OCA7) and showed the loss of biochemical interaction with Commander but not with Rab32. Finally, the authors attempted to illustrate that the Rab32-LRMDA-Commander possibly regulate the melanosome biogenesis by modulating the acidity, which possibly alters the PMEL processing and TYRP1 trafficking to maturing melanosomes. However, the authors identified new and alternate Rab32-LRMDA-Commander complex, but the data they have conducted and presented does not illustrate a straightforward role in melanosome biogenesis. Please consider the suggestions below, which may improve their manuscript.

Major comments:

1. The authors of this manuscript have previously shown the functionality of the SNX17-Commander complex in regulating the several integral membrane proteins from EEs to the cell surface. In this manuscript, authors identified another alternate complex Rab32-LRMDA-Commander using bioinformatics analysis followed by biochemical assays performed in non-melanocyte cell line HEK293T. Since Rab32 has been shown to regulate melanosome biogenesis and the mutation in LRMDA possibly associated with OCA7, the author diverted their functional studies towards melanosome biogenesis. In the end, they have presented two models for this pathway:
 - (a) SNX17-Commander complex regulates the cargo sorting to the cell surface or maturing melanosome- which was not demonstrated by the authors at the molecular level in this manuscript. Moreover, the loss of function of this complex is predicted to alter all early endosomal domains and cargo sorting (including non-melanocytic cargo), which will automatically block the formation of stage II melanosomes. Has the author tested the cargo sorting of both melanocytic and non-melanocytic proteins in the SNX17/Commander depleted cells?
 - (b) Authors predicted that Rab32-LRMDA-Commander complex functions from melanosomes, where no data was presented illustrating the complex localization or cargo sorting mediated by this complex from melanosomes (they have added a point about VAMP7 trafficking in their discussion) or the complex functions in regulating the tubule formation.
 - (c) This reviewer predicts that Rab32-LRMDA-Commander may act as a redundant complex to the SNX17-Commander complex on early endosomal sorting rather than at SEs/REs.
2. Authors observed more PMEL processivity in the KO cells of Rab32/LRMDA/Commander and predicted a change in pH, possibly altering the trafficking of melanocyte cargo. In this case, authors should have tested the functionality of V-ATPase or trafficking of other ion-transporters, including OCA7 (Santiago paper), to maturing melanosomes.
3. The authors used non-melanocytic cell lines, such as HEK293 cells, for all their biochemical interaction studies and used melanocytes, such as MNT1, for phenotypic analysis. Note that the observed interactions between Rab32-LRMDA-Commander may not be relevant in the context of melanocytes. However, they may have individual gene effects (as the authors did a single KO each time) on melanocyte pigmentation. This reviewer predicts that altering the EE/SE endosomal sorting will definitely affect the melanosome biogenesis as similar to BLOC-1-deficient cells (Setty et al., 2007)
4. Authors tried to connect the LRMDA mutations associated with OCA7 to pigmentation. However, it is not clear whether these affect the OCA7 protein function or its trafficking.
5. The phenotypes observed in the KO cells of STX17-Commander or Rab32-LRMDA-Commander complexes are probably due to the loss of membrane domain organization at the EEs rather than the critical effect on melanosome sorting pathways; those occur downstream of EEs mostly at the SE/REs.
6. Authors speculate that Retriever cannot interact with both LRMDA and SNX17 simultaneously (line 148). Has the author

tested the competitive binding between LRMDA and SNX17 with Retriever?

7. Authors exploited the AlphaFold2 at multiple places to analyze/predict the interactions between the complexes. How did the authors estimate the model accuracy while modelling their interactions, especially in relation to the LRMDA-Commander complex?

8. Authors should note that deacidification at Stage II may be better for TYR activity but not at Stage I, where their generated KO cells possess.

Minor comments:

1. Authors have tested the LRMDA-Commander interaction with Rab32. What about Rab38? It has been shown that Rab38 and 32 compensate for each other function in melanocytes (Wasmeier et al., 2006 and Bultema et al., 2012).

2. Line 145: Authors overstated that "LRMDA coupling to retriever is sufficient to recruit the entire Commander super-assembly"- authors only observed the interaction but did not show their localization to endosomal compartments.

3. Line 169: authors showed Rab32 binding with GFP-LRMDA and LRMD-GFP – which one is functional?

4. Line 235: The author interpreted that LRMDA acts as a direct linker between Rab32 and Retriever. This reviewer predicts that LRMDA may be a specific effector of Rab32. Since the interaction between LRMDA-Retriever possibly occurs on the membrane, where does Rab32 possibly facilitate the process? Consistently, authors showed the LRMDA (1-176) interaction with Rab32. Another possibility is that Rab32 facilitate the interaction between LRMDA and the Retriever on the membrane.

5. Legend for Fig. 3E is missing.

6. The graph in Fig. 4 E: The error bars are very dispersed, and it will be difficult to conclude this data.

7. Line 316: The title (LRMDA-dependent and independent) is very broad, and their data did not support any of these aspects.

8. Line 321: Melanosome maturation can be divided into "several" stages – it is very well established that it occurs in 'four' stages

9. Figure 5D: VPS35L KO cells showed enlarged endosomes, similar to BLOC-1-deficient melanocytes (Setty et al., 2007). But LRMDA KO cells contain more stage III melanosomes- Data suggests that VPS35L KO has a stronger effect than LRMDA, which will be an upstream factor to VPS35L in the proposed pathway.

10. Authors have used visual and spectrometric-based analysis of melanocyte pigmentation in their KO cells. They have not shown the loss of core pigment genes (TYRP1 and TYR) in the cells.

11. Authors have probed the processivity of PEML using the antibody. However, they have not purified the fibres biochemically (Nag et al., 2018).

12. Line 426: The mild increase in pigmentation with Bafilomycin A1 will tell us the defective V-ATPase, which should affect the lysosome-directed cargo such as EGFR degradation. In this condition, have the authors measured the recycling of any other cargo to PM to see whether the rescue has indeed occurred or if it only enhanced TYR activity in the immature melanosomes rather than improved PMEL processivity?

13. Line 481: overstated that SNX17-dependent and LRMDA-dependent steps regulate organelle biogenesis pathway. These complexes are predicted to have an effect due to defective endosomal sorting at the initial stage of maturation.

14. Line 506: authors placed the Rab32-LRMDA-Retriever on melanosome membrane- how do these molecules regulate the melanosome maturation?

Reviewer #3

(Remarks to the Author)

Butkovic et al. have identified the RAB32–LRMDA–Commander complex, demonstrating that LRMDA binds to the Commander complex via VPS35L, competing with SNX17 for the same binding site. Crucially, they show that this LRMDA–Commander interaction is essential for RAB32-dependent melanosome biogenesis. Furthermore, they reveal that the SNX17-Commander complex also contributes to melanosome formation. This study provides exciting insights into new biological functions of the Commander complex and elucidates the molecular basis of LRMDA mutations in oculocutaneous albinism type 7 (OCA7), which disrupt the interaction of LRMDA with either RAB32 or Commander. This manuscript makes a significant and valuable contribution to the field, with well-supported conclusions and the impact required for publication in this journal. However, a few issues should be addressed before publication.

The authors demonstrate that in HEK293T cells deletion of the N-terminal region clearly diminishes RAB32 binding but

does not affect the binding of any Retriever subunits or CCC complex subunits (Figure 2C). However, in HeLa cells, deletion of the N-terminal region significantly impacts VPS26C binding in (Supplementary Figure 2D). The authors should clarify the reasons for these discrepancies between cell lines.

With the deletion of the LRMDA C-terminal region, both in the ΔC construct (Figure 2C) and the 208* mutant (Figure 4E), an increase in LRMDA binding to RAB32 is observed. Could the authors provide a mechanistic explanation for this unexpected increase in binding affinity?

The authors should deposit or provide as a supplementary file the AlphaFold models used in this study.

The proposed LRMDA-Retriever interaction mechanism is based on an AlphaFold model. The authors have satisfactorily validated the interaction region between the LRMDA C-tail and VPS35L. Another interaction site involving Leu189 of LRMDA and VPS26C is also predicted. The authors should specify which residues of VPS26C are involved in this interaction and whether this region is evolutionarily conserved. If feasible, site-directed mutagenesis of VPS26C could further validate this binding site.

Line 357. The authors should clarify why VPS26C was used instead of VPS35L to identify proteins interacting with Retriever.

Issues with Figures and Figure Legends

- Figure 1E: In this volcano plot, a linear regression is used, whereas in the other volcano plots throughout the manuscript (4A-B, 6A, Supplementary Figure 3A), a paired t-test is employed. The authors should justify this methodological choice.
- Figure 3A & Supplementary Figure 1C: The labels for residues T39 and Q85 are positioned too far from their respective amino acid residues.
- Figure 3A: It is unclear what the zoomed-in and rotated (120°) inset is intended to illustrate. The figure title suggests that it highlights the interactions between LRMDA and RAB32, but the image does not clearly show the involvement of residues T39 and Q85 in this interaction. Since in lines 213–214 is mentioned that these residues are involved in the activation-inactivation states, it may be helpful to clarify this in the figure legend.
- Figure 3 Legend, Line 689: The panel designation should be "B" instead of "F," and the panel references from "B" to "D" do not match the actual figures.
- Figure 4D: The distinction between white LRMDA and gray RAB32 is unclear. Consider using light pink or another contrasting color for LRMDA.
- Figure 4G: The figure legend is missing.
- Figure 4F-H. The nomenclature should be consistent, either "C" or "cl.". This inconsistency also appears in the subsequent figures.
- Figure 4K: Although the color intensity of the solubilized melanin clearly indicates that the F145D mutant produces less melanin than the control, this difference is not reflected in the statistical analysis. However, the text (lines 310–311) states that it is. Could increasing the sample size help make this difference more evident?
- Figure 6A: In the MNT1 melanogenic cell line, the interactome analysis identifies a set of proteins with an increased fold change. However, two additional proteins with a decreased fold change are also visible. Additionally, two data points with identical values appear in Supplementary Figure 3A. Could the authors specify which proteins these are?
- Supplementary Figure 1A: It is difficult to distinguish the subunits in the Consurf-colored model. Indicating in the figure legend that the orientation matches Figure 2A would be helpful.
- Supplementary Figure 3B: The figure legend does not mention co-transfection. It only describes the transfection of GFP, GFP-RAB32, etc., but does not mention mCherry-LRMDA.
- Supplementary Raw Blots for Figures 2D and 2E: The raw blots do not correspond to the main text. Specifically, the raw blot 2D corresponds to Figure 2E, and vice versa.

References

- Singla et al. (2024) also identified LRMDA as a ligand of Retriever, using a similar approach. Their methodology involved defining a consensus motif based on their Retriever-SNX17 cryo-EM structure and mutagenesis studies, followed by a search for human proteins containing this conserved C-terminal motif. They further demonstrated robust *in vitro* binding of LRMDA to Retriever and confirmed that the LRMDA C-terminal tail interacts with the same VPS35L/VPS26C interface as SNX17. The authors should cite this study and discuss its relation to their findings.
- Line 107: The authors state that the motif is based on their previous structural and functional analyses. It would be beneficial to include the relevant citations here to clarify which studies are being referenced.

Discussion

Are there clinical differences between patients with LRMDA mutations that disrupt RAB32 binding (N117K) versus those affecting Retriever binding (Y208*, G217S, and R222*)? If they exhibit similar severity levels, this would further support the proposed mechanistic basis of OCA7.

Methods

Line 1063. Could the authors explain why GFP-Trap beads used for pulling down GFP-SNX17 were washed with water instead of buffer, as was done for the other samples?

Typos

- Line 70: "Leu-470" should be "Leu470" to maintain consistency with the amino acid nomenclature used throughout the manuscript.

- Line 970: "GFX PCR dNA" should be "GFX PCR DNA" (DNA in uppercase).
- Line 979: "sanger sequencing" should be "Sanger sequencing" (S in uppercase).
- Line 991: "Dnpl digestion" should be "DpnI digestion".
- Line 994: "muattion" should be "mutation".
- Line 1038. "HEES" should be "HEPES".
- In the Methods section there are several places where the spacing between numbers and units is inconsistent. It is recommended to review and standardize these spacings throughout the section.

Version 1:

Reviewer comments:

Reviewer #1

(Remarks to the Author)

The authors have satisfactorily addressed most of the reviewers' concerns. Point 3 regarding proteomic data deposition is incomplete. While the author's intention is clear, this is not complete until a valid PRIDE accession is included in the manuscript or supplementary information. Final submission should verify this is done. For point 4 regarding TMT proteomics methods details, the authors provided all key methodological parameters requested, except for specifying the labeling scheme (e.g., channels used per condition).

Reviewer #2

(Remarks to the Author)

In the revised manuscript, the authors answered my concerns and included the majority of the suggestions. I recommend the article for publication in the journal.

Reviewer #3

(Remarks to the Author)

The authors have adequately addressed all of the concerns raised by the reviewers. I would like to congratulate the authors for their fine work!

Reviewer #1 (Remarks to the Author):

This manuscript identified LRMDA as a novel adaptor protein that links the Commander complex to active RAB32, elucidating a previously unrecognised pathway essential for melanosome biogenesis and pigmentation. The study provides mechanistic insights into the molecular basis of oculocutaneous albinism type 7 (OCA7), demonstrating how disease-associated mutations in LRMDA disrupt its interaction with RAB32 and the Commander complex. By revealing the mutually exclusive nature of LRMDA-Commander and SNX17-Commander assemblies, the authors highlight the functional plasticity of Commander beyond its canonical roles in endosomal recycling. The study establishes that LRMDA functions independently of SNX17, indicating distinct roles for these assemblies in organelle biogenesis pathways. Overall, the manuscript is well written and presents solid proteomic evidence for LRMDA's role in Commander assembly and melanosome biogenesis. Minor points are listed below, which should be addressed before this manuscript can be accepted for publication.

We thank the reviewer for their positive and supportive comments.

1. RAB32 and RAB38 are regulators of intracellular trafficking, influencing various physiological processes, including pigmentation. Although the authors did not observe interactions between RAB38 and LRMDA (Figure 1E), the close structural similarity of RAB38 to RAB32 suggests potential redundancy or compensatory mechanisms. Can the authors indicate if differences in protein expression account for these observations and if these mechanisms may be active in cell types beyond RPE1?

We agree with the reviewer's comment suggesting that differences in expression might account for the lack of detected RAB38 in LRMDA interactome. Additionally, it is likely that LRMDA associates with RAB38 with lower affinity, as this would be consistent with the yeast two-hybrid screen performed in Beyers et al. (2022, JBC; PMID: 36334630).

We do detect RAB38 in the interactome of LRMDA and VPS26C expressed in MNT1 cells, however, the enrichment is low:

- GFP-LRMDA over GFP: RAB38 detected in 3/3 replicates, Log2FC = 0.94, p-value = 0.01
- GFP-VPS26C over GFP: RAB38 detected in 3/3 replicates, Log2FC = 0.68, p-value = 0.04

To illustrate this, in revised Figure 6A and Supplementary Figure 4A (previously Supplementary Figure 3A) we have labelled RAB38 in the volcano plots. In addition, we have performed immunoprecipitation of GFP-LRMDA expressed in MNT1 cells and probed for the presence of endogenous RAB38 comparing with endogenous RAB32. In a new Supplementary Figure 1H, we do observe an association with RAB38, binding that is lost with the LRMDA(F145D) mutation. Finally, we have added a discussion point to the manuscript text to cover the relationship with RAB38 (see lines: 224-227; 528-531).

2. In the section 'Cooperativity in binding within the RAB32:LRMDA:Retriever complex', the last statement regarding the lack of mechanistic explanation may unintentionally downplay the significance of the proteomic findings. Expanding on this in the discussion would provide valuable insights. The observed cooperativity in binding between these proteins likely arises from at least three synergistic mechanisms: 1) LRMDA functions as an adaptor protein, linking the Retriever complex to RAB32. The carboxy-terminal domain of LRMDA directly associates with the Retriever complex via a conserved motif (IRDDQL226), specifically binding to VPS35L; 2) In its active GTP-bound state RAB32 favors its interaction with LRMDA, and facilitate the recruitment of the Commander complex to the RAB32-decorated membrane; 3) Mutual enhancement of binding: The association between LRMDA and Retriever enhances the binding of LRMDA to RAB32, while conversely, RAB32 binding strengthens LRMDA's association with the Retriever complex. This suggests a positive feedback loop, where the binding of one partner promotes the interaction of the others. This cooperativity likely ensures the formation of a stable and functional RAB32:LRMDA:Retriever complex.

We thank the reviewer for the suggested explanation. We have included new text to clarify the likely significance of the cooperativity between RAB32, LRMDA and Commander binding in the manuscript (see lines: 278-280).

3. For the proteomic data, the authors should deposit their raw data files in a publicly accessible repository, like PRIDE, to ensure transparency and reproducibility of their findings, allowing other scientists to access and analyse the raw data used in the study.

The submission of proteomic data is ongoing and the accession number of deposited data will be indicated in the text.

4. In the method section, the authors should specify the experimental workflow used in the tandem mass tag (TMT)-based quantitative proteomics, including the TMT labeling scheme, criteria for peptide and protein identification. They should clarify if all data reported for quantitation used SPS-MS3 and indicate mass tolerance applied to fragment ions (e.g. fragment ion tolerance was set to 0.9 Da for SPS-MS3). More information is required to assess how authors filtered PSMs with poor quality (e.g. spectra with TMT reporter summed signal-to-noise measurements that were less than 200, or with no MS3 spectra were excluded from quantitation).

We have added additional details to the Methods section (see lines: 1206-1208, 1225-1227).

Yes, all data reported for quantitation used SPS-MS3. To clarify, the MS2 spectra were used for peptide identification ('MS/MS tolerance was set at 0.6Da'), and the MS3 spectra were used for reporter ion detection, which was set at 20 ppm in the Reporter Ions Quantifier node of the Proteome Discoverer software.

Spectra were excluded from quantitation if the percentage of ion current in the isolation window not originating from the identified peptide was >75%; if the average

reporter S/N for the spectra was <10 or is the SPS masses matching to the identified peptide was <65%.

5. The specific statistical methods employed for data analysis is not consistent between method and figure caption. For example, in the method they indicated p-value adjusted using the Benjamini-Hochberg FDR method, but Figure 1E only indicates significantly enriched proteins with $p < 0.01$ and Log2 fold change > 2 are labelled, which appear arbitrarily defined. Clarification is also needed on the number of biological and technical replicates included to assess reproducibility and statistical power. For example, Figure 1E stated $n=6$, but there is no indication if these are biological or technical replicates.

We have added additional information to the figure legends and methods section.

As is conventional with the proteomic analysis, the Log2FC and p-value cut off values are arbitrary, and we used stringent cut off values to reduce identification of false-positives, prioritise biological relevance and enable confirmation of changes through follow-up analyses

The p-value threshold of 0.01 that was used in the volcano plots is more stringent than the conventional 0.05 used by many studies, as is the effect size cut-off of a Log2 fold change of >2 . Whilst we accept that it would be ideal to utilize an FDR-stringency cut-off, it is commonly understood that in proteomics analyses, FDR cut-offs are useful, but not essential as they are typically overly stringent and can result in false-negatives (Pascovici et al., 2016, Proteomics, PMID: 27461997). To provide information regarding FDR, we have now revised the volcano plots and depicted all points with $FDR < 0.05$ as triangles.

6. Proteomics analyses identified LRMDA to accession #: A0A087WWI0 (226 AA) though another similar variant exists: Q9H2I8 (198 AA). Can the authors clarify if their mass spectrometry data allowed differentiation between these variants?

In our proteomic analysis, we detected unique peptides specific to the longer isoform, along with peptides shared between the shorter and longer isoforms. Because the shorter isoform lacks any unique peptides, the unique peptides belonging to LRMDA were integrated under the common accession A0A087WWI0.

Moreover, we have observed when trying to express the shorter isoform *in vitro*, that it lacks stability compared with the longer isoform. Together, we believe therefore that the longer form is more functionally relevant despite the better annotation of the shorter isoform in the UniProt database.

Reviewer #2 (Remarks to the Author):

Comments to the author:

Melanosome biogenesis follows sequential maturation of endocytic organelles into mature melanosomes in four stages. This process requires sequential transport of melanocytic cargo to form these unique organelles. Sorting endosomes (SE) and

recycling endosomes (RE) play key roles during these processes. However, the role of early endosomal (EE) cargo sorting pathways in contributing the melanosome maturation is unknown. In this manuscript, the authors tested the role of one such sorting mechanism involving the Commander complex. The authors initiated their studies by identifying the LRMDA (leucine-rich melanocyte differentiation-associated) protein as an interactor to the Commander complex. Their in vitro modelling studies illustrated that LRMDA binds weakly (compared to SNX17) to the Commander complex in association with active Rab32.

We wish to clarify that our study does not contain data quantifying the affinities of SNX17 and LRMDA for binding to Retriever. We, therefore, make no conclusions as to their comparative binding affinities.

They used biochemical assays to prove the interaction between Rab32-LRMDA-Commander. Moreover, they have carried out gene knockouts (KO) in melanocyte cell line MNT1 and showed a decrease in melanocyte pigmentation. Moreover, the authors used pathogenic mutants of LRMDA that are associated with oculocutaneous albinism type 7 (OCA7) and showed the loss of biochemical interaction with Commander but not with Rab32.

To clarify, we establish that the OCA7-causative mutant, LRMDA(N117K), displays decreased binding to RAB32 (see Figure 4E).

Finally, the authors attempted to illustrate that the Rab32-LRMDA-Commander possibly regulate the melanosome biogenesis by modulating the acidity, which possibly alters the PMEL processing and TYRP1 trafficking to maturing melanosomes. However, the authors identified new and alternate Rab32-LRMDA-Commander complex, but the data they have conducted and presented does not illustrate a straightforward role in melanosome biogenesis. Please consider the suggestions below, which may improve their manuscript.

Major comments:

1. The authors of this manuscript have previously shown the functionality of the SNX17-Commander complex in regulating the several integral membrane proteins from EEs to the cell surface. In this manuscript, authors identified another alternate complex Rab32-LRMDA-Commander using bioinformatics analysis followed by biochemical assays performed in non-melanocyte cell line HEK293T. Since Rab32 has been shown to regulate melanosome biogenesis and the mutation in LRMDA possibly associated with OCA7, the author diverted their functional studies towards melanosome biogenesis. In the end, they have presented two models for this pathway:

(a) SNX17-Commander complex regulates the cargo sorting to the cell surface or maturing melanosome- which was not demonstrated by the authors at the molecular level in this manuscript. Moreover, the loss of function of this complex is predicted to alter all early endosomal domains and cargo sorting (including non-melanocytic cargo), which will automatically block the formation of stage II melanosomes. Has the author tested the cargo sorting of both melanocytic and non-melanocytic proteins in the SNX17/Commander depleted cells?

In the original manuscript, we show that SNX17 KO cells display a defect in trafficking of the melanocytic TYRP1 protein as evidenced by increased colocalization with the late endosome/lysosome marker LAMP1 (see Figure 7A). We have now performed new cell surface biotinylation assays to establish that SNX17 KO in MNT1 cells leads to the well characterised perturbation in the recycling of the non-melanocytic protein LRP1. We have established that this phenotype is also observed in individual VPS35L (Retriever), CCDC22 (CCC complex) and FAM21 (WASH complex) KO MNT1 cell lines, but is NOT observed in RAB32 or LRMDA KO MNT1 cell lines (see Supplementary Figure 4C).

(b) Authors predicted that Rab32-LRMDA-Commander complex functions from melanosomes, where no data was presented illustrating the complex localisation or cargo sorting mediated by this complex from melanosomes (they have added a point about VAMP7 trafficking in their discussion) or the complex functions in regulating the tubule formation.

Our model is supported by the publication that detected COMMD3 on melanosomes (Bajpai et al, Science 2023, Figure 5A) and previous knowledge that RAB32 and LRMDA localize to melanosomes. In new data we have rescued LRMDA KO cells with wild type mCherry-LRMDA or mCherry-LRMDA(L226G) (defective in Commander binding) and observed by spinning disk microscopy colocalization with GFP-VPS35L. This has revealed a partial co-localisation of mCherry-LRMDA with GFP-VPS35L, that is lost in the mCherry-LRMDA(L226G) cells. These data are included in Supplementary Figure 3B.

(c) This reviewer predicts that Rab32-LRMDA-Commander may act as a redundant complex to the SNX17-Commander complex on early endosomal sorting rather than at SEs/REs.

Our functional data argue that these two complexes are not redundant, as knock-out of LRMDA generates distinct phenotypes compared with knock-out of SNX17 within the context of melanogenic enzyme trafficking, PMEL processing, compartment neutralisation and actin dynamics (see Figure 7), and in the endosomal recycling of LRP1 (Supplementary Figure 4C). The latter is consistent with SNX17's well-characterised role in directly binding to NPxY/NxxY sorting motifs present in cargo proteins (including LRP1) to initiate their sorting and recycling. We have no data suggestive of LRMDA directly binding to cargo and therefore having a direct role in sequence-dependent cargo sorting. Conceptually therefore SNX17 and LRMDA have distinct biochemical features within the context of cargo sorting.

2. Authors observed more PMEL processivity in the KO cells of Rab32/LRMDA/Commander and predicted a change in pH, possibly altering the trafficking of melanocyte cargo. In this case, authors should have tested the functionality of V-ATPase or trafficking of other ion-transporters, including OCA7 (Santiago paper), to maturing melanosomes.

To clarify, OCA7/LRMDA is a peripheral protein that has no structural features consistent with it functioning as an ion transporter.

3. The authors used non-melanocytic cell lines, such as HEK293 cells, for all their

biochemical interaction studies and used melanocytes, such as MNT1, for phenotypic analysis. Note that the observed interactions between Rab32-LRMDA-Commander may not be relevant in the context of melanocytes. However, they may have individual gene effects (as the authors did a single KO each time) on melanocyte pigmentation.

We would like to note that our functional assays establish that knock-out of LRMDA or Commander subunits lead to pigmentation defects in melanocytes (see Figures 4J-K, 5B-D, 6C-D, 7D-E). Moreover, by performing the rescue experiments using LRMDA mutants specifically disrupting RAB32 or Commander binding, we show that they fail to rescue the pigmentation defect (see Figure 4I-K). These data indicate that the assembly with RAB32 and Commander is required for the function of LRMDA in pigmentation. In addition, we demonstrate that the OCA7-causative mutations perturb the assembly of RAB32-LRMDA-Commander.

Altogether, we consider that our data provides functional context for this assembly in melanocytes.

This reviewer predicts that altering the EE/SE endosomal sorting will definitely affect the melanosome biogenesis as similar to BLOC-1-deficient cells (Setty et al., 2007).

We agree and indeed included statements to this effect within the original manuscript. However, we have now clarified that the function of Commander in the organisation of endosomal subdomains (Butkovic et al., 2024, Nat Commun., PMID: 39168982) may broadly affect trafficking from endosomes (lines 584-588).

4. Authors tried to connect the LRMDA mutations associated with OCA7 to pigmentation. However, it is not clear whether these affect the OCA7 protein function or its trafficking.

As to function, we clearly define that in OCA7 the protein function is perturbed with respect to binding to RAB32 and Commander (see Figure 4E) and that in rescue experiments within a LRMDA KO background, mutants unable to associate with either RAB32 or Commander fail to rescue the pigmentation defect (see Figure 4I-K). Regarding the trafficking issue, we would like to clarify that OCA7/LRMDA is a peripheral cytosolic protein and hence its trafficking via classic membrane transport pathways is not relevant.

5. The phenotypes observed in the KO cells of STX17-Commander or Rab32-LRMDA-Commander complexes are probably due to the loss of membrane domain organisation at the EEs rather than the critical effect on melanosome sorting pathways; those occur downstream of EEs mostly at the SE/REs.

We do not disagree with the reviewer that the knock-out of Commander subunits affects the organisation of endosomal subdomains, and indicate that Commander likely plays a more 'global' role in endosomal sorting (lines 462-463 in the original manuscript, added lines 584-588 in the updated manuscript). Indeed, a previous study conducted in our lab (Butkovic et al., 2024, Nat Commun., PMID: 39168982) has demonstrated that knock-out of VPS35L affects the distribution of different endosomal markers and likely perturbs endosomal subdomain organisation.

Importantly, the same study demonstrated that this is independent of Commander associating with SNX17 (see Figure 6 and Supplementary Figure 4 in Butkovic et al., 2024, Nat Commun., PMID: 39168982). So, while we cannot exclude a role for SNX17 (we assume that STX17 is a typo) in subdomain organisation in early endosomes, specifically within melanocytes there are no data to support such conclusion. Furthermore, a role for LRMDA in the organisation of early endosomal sorting subdomains is unlikely, as previous research (Beyers et al., 2022, JBC; PMID: 36334630) has indicated that LRMDA does not co-localise with early endosomal markers.

6. Authors speculate that Retriever cannot interact with both LRMDA and SNX17 simultaneously (line 148). Has the author tested the competitive binding between LRMDA and SNX17 with Retriever?

We thank the reviewer for this suggestion. We have not thus far tested the competitive binding between LRMDA and SNX17. However, based on the predicted and validated AlphaFold modelling of the shared binding interface of SNX17 and LRMDA to VPS35L we predict that their affinities will be comparable. Presently, we are performing supported bilayer-based reconstitution of RAB32-LRMDA-Commander that will examine this question. We anticipate that these data, along with a defined mechanisms for the autoinhibition/regulator interactions and a cryoET structure of the membrane assembled complex will form part of a future manuscript.

7. Authors exploited the AlphaFold2 at multiple places to analyse/predict the interactions between the complexes. How did the authors estimate the model accuracy while modelling their interactions, especially in relation to the LRMDA-Commander complex?

We have assessed the quality of predictions across all 5 models generated for each prediction by assessing pLDDT scores, PAE values and iPTM scores, and have supplied the pLDDT (model coloured by pLDDT scores) and PAE information in Supplementary Figure 4. Moreover, we analysed the principles of RAB32-LRMDA-Commander assembly by generating truncations in LRMDA (see Figure 2C), and tested the directness of the interactions through in vitro assay (see Figure 3E). Furthermore, for key interactions we have validated the models through site-directed mutagenesis (see Figures 1D, 2D, 2E, 3B, 3C).

8. Authors should note that deacidification at Stage II may be better for TYR activity but not at Stage I, where their generated KO cells possess.

We thank the reviewer for this comment but note that Stage II melanosomes are not expected to contain significant levels of TYR. The arrival of TYR (and other cargo) is what is believed to define the transition from Stage II to Stage III melanosomes.

Minor comments:

1. Authors have tested the LRMDA-Commander interaction with Rab32. What about Rab38? It has been shown that Rab38 and 32 compensate for each other function in melanocytes (Wasmeier et al., 2006 and Bultema et al., 2012).

We thank the reviewer for highlighting RAB38 and refer to our earlier comment for Reviewer 1 Point 1, which we reproduce here for clarity:

We agree with the reviewer's comment suggesting that differences in expression might account for the lack of detected RAB38 in LRMDA interactome. Additionally, it is likely that LRMDA associates with RAB38 with lower affinity, as this would be consistent with the yeast two-hybrid screen performed in Beyers et al. (2022, JBC; PMID: 36334630).

We do detect RAB38 in the interactome of LRMDA and VPS26C expressed in MNT1 cells, however, the enrichment is low:

- *GFP-LRMDA over GFP: RAB38 detected in 3/3 replicates, Log2FC = 0.94, p-value = 0.01*
- *GFP-VPS26C over GFP: RAB38 detected in 3/3 replicates, Log2FC = 0.68, p-value = 0.04*

To illustrate this, in revised Figure 6A and Supplementary Figure 4A (previously Supplementary Figure 3A) we have labelled RAB38 in the volcano plots. In addition, we have performed immunoprecipitation of GFP-LRMDA expressed in MNT1 cells and probed for the presence of endogenous RAB38 comparing with endogenous RAB32. In a new Supplementary Figure 1H, we do observe an association with RAB38, binding that is lost with the LRMDA(F145D) mutation. Finally, we have added a discussion point to the manuscript text to cover the relationship with RAB38 (see lines: 224-226; 528-531).

2. Line 145: Authors overstated that "LRMDA coupling to retriever is sufficient to recruit the entire Commander super-assembly"- authors only observed the interaction but did not show their localisation to endosomal compartments.

See comment to this Reviewer Major Comment 1B and the reference to new imaging data in Supplementary Figure 3B. We have also revised our comment to read: "LRMDA coupling to Retriever is required to engage....."

3. Line 169: authors showed Rab32 binding with GFP-LRMDA and LRMD-GFP – which one is functional?

By defining the precise mechanism of LRMDA association with Commander we have shown the essential role of the carboxy-terminus of LRMDA. The LRMDA-GFP fusion, which occludes the Commander binding tail with the GFP tag, fails to associate with Commander (see data in Figure 1D). Therefore, while both GFP-LRMDA and LRMDA-GFP can bind to RAB32, we only used GFP-tagged LRMDA for all rescue experiments.

4. Line 235: The author interpreted that LRMDA acts as a direct linker between Rab32 and Retriever. This reviewer predicts that LRMDA may be a specific effector of Rab32. Since the interaction between LRMDA-Retriever possibly occurs on the membrane, where does Rab32 possibly facilitate the process? Consistently, authors showed the LRMDA (1-176) interaction with Rab32. Another possibility is that Rab32 facilitate the interaction between LRMDA and the Retriever on the membrane.

Our in vitro reconstitution using purified proteins establishes that RAB32 does not directly associate with Retriever but requires LRMDA to act as the molecular 'bridge' through forming the RAB32:LRMDA:Retriever assembly. We do not disagree with the comment that '*Rab32 facilitate the interaction between LRMDA and the Retriever on the membranes*'. A core message of our original manuscript is that we propose that by binding to LRMDA, Commander can associate with RAB32-decorated membranes (for example, see lines 239-241 and 331-333 and the final figure).

5. Legend for Fig. 3E is missing.

Apologies this has been corrected.

6. The graph in Fig. 4 E: The error bars are very dispersed, and it will be difficult to conclude this data.

We thank the reviewer for this observation. The data dispersion is likely a consequence of normalisation to GFP levels with some mutants expressing at slightly lower levels compared to the wild-type GFP-LRMDA. We would like to note, however, that the near-complete loss of binding to RAB32 (N117K) or Commander subunits (Y208*, Q217S and R222*) has been consistent across the three replicates and those data are not dispersed.

7. Line 316: The title (LRMDA-dependent and independent) is very broad, and their data did not support any of these aspects.

We have corrected this title which now reads "*Retriever function is required for melanosome biogenesis*".

8. Line 321: Melanosome maturation can be divided into "several" stages – it is very well established that it occurs in 'four' stages.

This has been corrected in the revised text.

9. Figure 5D: VPS35L KO cells showed enlarged endosomes, similar to BLOC-1-deficient melanocytes (Setty et al., 2007). But LRMDA KO cells contain more stage III melanosomes- Data suggests that VPS35L KO has a stronger effect than LRMDA, which will be an upstream factor to VPS35L in the proposed pathway.

We thank the reviewer for this observation and would like to note that we stated in the original manuscript that Commander likely plays several roles (LRMDA-dependent and LRMDA-independent) in melanosome biogenesis. Indeed, we propose that in addition to RAB32-LRMDA-Commander, the canonical SNX17-Commander assembly also plays a role in melanosome function (see Figures 6F-G, 7).

10. Authors have used visual and spectrometric-based analysis of melanocyte pigmentation in their KO cells. They have not shown the loss of core pigment genes (TYRP1 and TYR) in the cells.

By imaging we have shown that TYRP1 remains present in our KO cell lines, see Figure 7A of the original manuscript.

11. Authors have probed the processivity of PEML using the antibody. However, they have not purified the fibres biochemically (Nag et al., 2018).

Yes, we have not purified PMEL fibrils, and we believe that this is not necessary given the results shown in Figure 7B with RIPA buffer extracts.

12. Line 426: The mild increase in pigmentation with Bafilomycin A1 will tell us the defective V-ATPase, which should affect the lysosome-directed cargo such as EGFR degradation. In this condition, have the authors measured the recycling of any other cargo to PM to see whether the rescue has indeed occurred or if it only enhanced TYR activity in the immature melanosomes rather than improved PMEL processivity?

We thank the reviewer for this observation. We have now included in the text that the rescue might occur due to enhanced TYR activity in immature melanosomes (see lines 460-462).

13. Line 481: overstated that SNX17-dependent and LRMDA-dependent steps regulate organelle biogenesis pathway. These complexes are predicted to have an effect due to defective endosomal sorting at the initial stage of maturation.

We refer back to our reply to Point 5 of the Major Comments of this reviewer. In the discussion we have added text to state this reviewers' point of view (see lines 584-588).

14. Line 506: authors placed the Rab32-LRMDA-Retriever on melanosome membrane- how do these molecules regulate the melanosome maturation?

Beyond providing the evidence for the involvement of this assembly in the maturation pathway, we do not know how these proteins regulate melanosome maturation. This will require much more extensive functional analysis that is beyond the scope of the present study.

Reviewer #3 (Remarks to the Author):

Butkovic et al. have identified the RAB32–LRMDA–Commander complex, demonstrating that LRMDA binds to the Commander complex via VPS35L, competing with SNX17 for the same binding site. Crucially, they show that this LRMDA–Commander interaction is essential for RAB32-dependent melanosome biogenesis. Furthermore, they reveal that the SNX17-Commander complex also contributes to melanosome formation. This study provides exciting insights into new biological functions of the Commander complex and elucidates the molecular basis of LRMDA mutations in oculocutaneous albinism type 7 (OCA7), which disrupt the interaction of LRMDA with either RAB32 or Commander. This manuscript makes a significant and valuable contribution to the field, with well-supported conclusions and the impact required for publication in this journal. However, a few issues should be addressed before publication.

We thank the reviewer for their positive and supportive comments.

The authors demonstrate that in HEK293T cells deletion of the N-terminal region clearly diminishes RAB32 binding but does not affect the binding of any Retriever subunits or CCC complex subunits (Figure 2C). However, in HeLa cells, deletion of the N-terminal region significantly impacts VPS26C binding in (Supplementary Figure 2D). The authors should clarify the reasons for these discrepancies between cell lines.

We believe that this is a manifestation of the autoinhibition/regulation of RAB32 and LRMDA binding to Retriever (as discussed in the manuscript) that becomes apparent through the relative expression levels of the expressed transgene between these two cell lines. Given this complexity, we have chosen not to explore this phenomenon in more detail within a cellular context but have instead decided to establish a reconstitution system using recombinant proteins/complexes on supported lipid bilayers to precisely tease apart the complex mechanism of this autoinhibition/regulation. This work will form the basis of a future independent study.

With the deletion of the LRMDA C-terminal region, both in the ΔC construct (Figure 2C) and the 208 mutant (Figure 4E), an increase in LRMDA binding to RAB32 is observed. Could the authors provide a mechanistic explanation for this unexpected increase in binding affinity?*

We thank the reviewer for this observation, but unfortunately, we cannot provide a mechanistic explanation for this. We propose that it is likely that LRMDA is autoinhibited (or otherwise regulated), and by removing the carboxy-terminus, we can ameliorate such autoinhibition. However, we could not find a molecular mechanism for this effect on RAB32 binding.

The authors should deposit or provide as a supplementary file the AlphaFold models used in this study.

We have deposited the Retriever-LRMDA-RAB32 (ma-j7h7g), LRMDA-RAB32 (ma-2bezz) and LRMDA-RAB38 (ma-f3hs4) models onto ModelArchive, and these will be made publicly available following the publication of the manuscript.

The proposed LRMDA-Retriever interaction mechanism is based on an AlphaFold model. The authors have satisfactorily validated the interaction region between the LRMDA C-tail and VPS35L. Another interaction site involving Leu189 of LRMDA and VPS26C is also predicted. The authors should specify which residues of VPS26C are involved in this interaction and whether this region is evolutionarily conserved. If feasible, site-directed mutagenesis of VPS26C could further validate this binding site.

We have generated the necessary single site-directed mutants and established that they do not appear to have a major effect on the association with LRMDA. While these data may suggest that the AlphaFold predicted association with VPS26C is incorrect, they do serve to re-enforce that the terminal leucine motif is the major driver of LRMDA-Retriever association.

Line 357. The authors should clarify why VPS26C was used instead of VPS35L to identify proteins interacting with Retriever.

We utilised GFP-VPS26C for technical reasons as its expression was higher in melanocytes than when expressing the larger VPS35L-GFP.

Issues with Figures and Figure Legends

- Figure 1E: In this volcano plot, a linear regression is used, whereas in the other volcano plots throughout the manuscript (4A-B, 6A, Supplementary Figure 3A), a paired t-test is employed. The authors should justify this methodological choice.

We thank the reviewer for this comment. To make the analysis consistent, we have now used paired t-test to analyse the data depicted in Figure 1E and replaced the figure accordingly. The revised volcano plot still shows the significant enrichment of all Commander subunits and RAB32 in LRMDA interactome.

- Figure 3A & Supplementary Figure 1C: The labels for residues T39 and Q85 are positioned too far from their respective amino acid residues.

Corrected.

- Figure 3A: It is unclear what the zoomed-in and rotated (120°) inset is intended to illustrate. The figure title suggests that it highlights the interactions between LRMDA and RAB32, but the image does not clearly show the involvement of residues T39 and Q85 in this interaction. Since in lines 213–214 is mentioned that these residues are involved in the activation-inactivation states, it may be helpful to clarify this in the figure legend.

We have clarified this point within the figure legend.

- Figure 3 Legend, Line 689: The panel designation should be "B" instead of "F," and the panel references from "B" to "D" do not match the actual figures.

Corrected.

- Figure 4D: The distinction between white LRMDA and gray RAB32 is unclear. Consider using light pink or another contrasting color for LRMDA.

Corrected, we have now coloured RAB32 in blue.

- Figure 4G: The figure legend is missing. Corrected.

- Figure 4F-H. The nomenclature should be consistent, either "C" or "cl.". This inconsistency also appears in the subsequent figures.

Corrected.

- Figure 4K: Although the color intensity of the solubilised melanin clearly indicates that the F145D mutant produces less melanin than the control, this difference is not

reflected in the statistical analysis. However, the text (lines 310–311) states that it is. Could increasing the sample size help make this difference more evident?

While this is a valid point, we prefer to maintain consistency between data within the manuscript by defining statistical significance through the same number of biological repeats. We have modified the text to indicate that the change was not statistically significant.

- Figure 6A: In the MNT1 melanogenic cell line, the interactome analysis identifies a set of proteins with an increased fold change. However, two additional proteins with a decreased fold change are also visible. Additionally, two data points with identical values appear in Supplementary Figure 3A. Could the authors specify which proteins these are?

GFP and MICAL2, in both cases. GFP-VPS26C and GFP-LRMDA expressed at lower levels than GFP, a possible technical issue due to the larger size of chimeric proteins compared to the GFP tag.

- Supplementary Figure 1A: It is difficult to distinguish the subunits in the Consurf-colored model. Indicating in the figure legend that the orientation matches Figure 2A would be helpful.

Corrected.

- Supplementary Figure 3B: The figure legend does not mention co-transfection. It only describes the transfection of GFP, GFP-RAB32, etc., but does not mention mCherry-LRMDA.

Corrected.

- Supplementary Raw Blots for Figures 2D and 2E: The raw blots for do not correspond to the main text. Specifically, the raw blot 2D corresponds to Figure 2E, and vice versa.

Corrected.

References

- Singla et al. (2024) also identified LRMDA as a ligand of Retriever, using a similar approach. Their methodology involved defining a consensus motif based on their Retriever-SNX17 cryo-EM structure and mutagenesis studies, followed by a search for human proteins containing this conserved C-terminal motif. They further demonstrated robust in vitro binding of LRMDA to Retriever and confirmed that the LRMDA C-terminal tail interacts with the same VPS35L/VPS26C interface as SNX17. The authors should cite this study and discuss its relation to their findings.

Reference has been included.

- Line 107: The authors state that the motif is based on their previous structural and functional analyses. It would be beneficial to include the relevant citations here to clarify which studies are being referenced.

Reference has been included.

Discussion

Are there clinical differences between patients with LRMDA mutations that disrupt RAB32 binding (N117K) versus those affecting Retriever binding (Y208, G217S, and R222*)? If they exhibit similar severity levels, this would further support the proposed mechanistic basis of OCA7.*

There is very limited clinical data on N117K (in the literature this is referred to as N89K in the shorter LRMDA isoform). Additionally, we are hesitant to over-interpret those reported clinical data as the patient number is low, and we are using only one mutant that specifically loses RAB32 binding.

Methods

Line 1063. Could the authors explain why GFP-Trap beads used for pulling down GFP-SNX17 were washed with water instead of buffer, as was done for the other samples?

We thank the reviewer for this observation, the beads were washed in equivalent buffer, prepared in dH₂O, instead of PBS. The text has now been corrected.

Typos

- Line 70: "Leu-470" should be "Leu470" to maintain consistency with the amino acid nomenclature used throughout the manuscript.

This has been corrected.

- Line 970: "GFX PCR dNA" should be "GFX PCR DNA" (DNA in uppercase).

This has been corrected.

- Line 979: "sanger sequencing" should be "Sanger sequencing" (S in uppercase).

This has been corrected.

- Line 991: "Dnpl digestion" should be "DpnI digestion".

This has been corrected.

- Line 994: "muattion" should be "mutation".

This has been corrected.

- Line 1038. "HEES" should be "HEPES".

This has been corrected.

- In the Methods section there are several places where the spacing between numbers and units is inconsistent. It is recommended to review and standardize these spacings throughout the section.

This has been corrected.

RESPONSE TO REVIEWERS' COMMENTS

Reviewer #1 (Remarks to the Author):

The authors have satisfactorily addressed most of the reviewers' concerns. Point 3 regarding proteomic data deposition is incomplete. While the author's intention is clear, this is not complete until a valid PRIDE accession is included in the manuscript or supplementary information. Final submission should verify this is done. For point 4 regarding TMT proteomics methods details, the authors provided all key methodological parameters requested, except for specifying the labeling scheme (e.g., channels used per condition).

We thank the reviewer 1 for their comments. We have now deposited the data onto the PRIDE database, and included the following statement in the manuscript: The mass spectrometry proteomics data have been deposited to the ProteomeXchange Consortium via the PRIDE [75] partner repository with the dataset identifier PXD067193.

Reviewer #2 (Remarks to the Author):

In the revised manuscript, the authors answered my concerns and included the majority of the suggestions. I recommend the article for publication in the journal.

We thank the reviewer for their assessment of our manuscript.

Reviewer #3 (Remarks to the Author):

The authors have adequately addressed all of the concerns raised by the reviewers. I would like to congratulate the authors for their fine work!

We thank the reviewer for their comments.